# Temperature-driven mechanistic transition in propylene oxidation over Pt/CeO$_2$ ensemble catalysts

Zihao Li[1,2], Xingyan Chen[1], Yao Lv[3], Sheng Dai [3], Huazhen Chang[4], Zhenguo Li[5], Kailong Ye[6], Fudong Liu [6] ✉, Lei Ma [1] ✉ & Naiqiang Yan [1]

Pt/CeO$_2$ ensemble catalysts are promising for propylene (C$_3$H$_6$) oxidation in vehicle exhaust, yet identifying the intrinsic active sites and understanding how the metal-support interface evolves at varying reaction temperatures remains contentious. Herein, we demonstrate that H$_2$-activated Pt/CeO$_2$ ensemble catalysts feature metallic Pt ensembles as intrinsic active sites, lowering the 50% conversion temperature by 120 °C after hydrogen activation. Various operando characterization techniques reveal an approximately 170 °C threshold temperature for the dynamic change of the reaction models. Meanwhile, kinetics and theoretical analysis illustrates that oxygen-facilitated dehydrogenation of $sp^3$ C-H bonds is the rate-determining step. At low temperatures, both C$_3$H$_6$ and O$_2$ adsorb and activate on metallic Pt, without CeO$_2$ involvement. Once the temperature exceeds threshold, C$_3$H$_6$ fully covers Pt sites, while O$_2$ activates over Pt-O-Ce interfaces and participates in dehydrogenation. This study highlights the dynamic nature of oxygen activation, leading to distinct reaction temperature regimes during C$_3$H$_6$ oxidation.

Pt is recognized as one of the most active components for diesel oxidation catalysts and three-way catalysts in vehicle emission control[1,2]. CeO$_2$ has been commonly used as the supporting materials for Pt-based emission control catalysts, due to the decent oxygen storage capacity and excellent reducibility[1]. Wherein, the defect sites and sufficient active oxygen species at the interfacial sites contributed to anchoring Pt active species and promoting low-temperature oxidation activity[3,4]. In the past decade, atomically dispersed Pt catalysts received much attention owing to the maximum utilization of precious metals. Especially, Pt single-atom catalysts (Pt SACs) exhibited outstanding reactivity towards CO oxidation[5,6]. However, the absence of Pt ensemble sites (Pt$_e$) still constrains the catalytic activity towards alkene (such as C$_3$H$_6$) oxidation[7]. Therefore, great efforts have been devoted to fabricating Pt ensemble catalysts that could effectively accomplish the cleavage of C–H methyl or C = C double bonds during C$_3$H$_6$ oxidation[8,9]. It is still challenging to elucidate the intrinsic active species of Pt$_e$/CeO$_2$ ensemble catalysts and clarify the specific reaction pathways for the catalytic oxidation of C$_3$H$_6$.

The diversity in valence states (electronic properties) was one of the decisive aspects determining the reactivity of Pt$_e$ catalysts, besides considering the geometry and size as critical factors. So far, it remains debated the oxidation state in charge of the intrinsic activity of atomically dispersed Pt catalysts. It was reported that Pt$^0$ or Pt$^{\delta+}$ were the intrinsic active sites for catalytic oxidation of C$_3$H$_6$ from the

[1]State Key Laboratory of Green Papermaking and Resource Recycling, School of Environmental Science and Engineering, Shanghai Jiao Tong University, Shanghai, China. [2]PetroChina Petrochemical Research Institute, CNPC Company, Beijing, China. [3]Key Laboratory for Advanced Materials and Feringa Nobel Prize Scientist Joint Research Center, School of Chemistry & Molecular Engineering, East China University of Science and Technology, Shanghai, China. [4]School of Chemistry and Life Resources, Renmin University of China, Beijing, China. [5]National Engineering Laboratory for Mobile Source Emission Control Technology, China Automotive Technology & Research Center Co. Ltd., Tianjin, China. [6]Department of Chemical and Environmental Engineering, Bourns College of Engineering, Center for Environmental Research and Technology (CE-CERT), Materials Science and Engineering (MSE) Program, UCR Center for Catalysis, University of California, Riverside, CA, USA. ✉e-mail: fudong.liu@ucr.edu; leima8@sjtu.edu.cn

previous literature[10–12], which might be strongly affected by the catalyst support and promoters. The contentious issue of the oxidation state of Pt catalysts has also been investigated with CO oxidation, with intrinsic similarity to $C_3H_6$ oxidation. For example, Maurer et al. indicated that $Pt_X^{\delta+}$ forming during catalytic oxidation of CO were the exclusive active sites[13]. Ding et al., however, discovered that $Pt^0$ was the sole active phase and $Pt^{\delta+}$ was the spectator for catalytic oxidation of CO[14]. Therefore, complementary investigation of intrinsic $Pt_e$ species with specific electronic properties could provide a guide for the rational design of highly efficient $Pt_e$ catalysts.

Regarding the reaction pathway of catalytic oxidation of $C_3H_6$, either the reactive oxygen species (superoxide and peroxide) transformed from gaseous $O_2$ or lattice oxygen from interfacial sites might engage in catalytic oxidation of $C_3H_6$. On one hand, $C_3H_6$ could adsorb over the catalyst surface, and the reaction took place between the $C_3H_6$ molecules and reactive oxygen species to form intermediates[10,15]. On the other hand, $C_3H_6$ oxidation might follow the Mars-van Krevelen mechanism, in which $C_3H_6$ oxidation could react with lattice oxygen, causing the appearance of anion vacancies, followed by the re-oxidation of catalysts by gaseous oxygen in a separate step[16]. Yet, it might be subjective to directly conclude the catalytic oxidation reaction model without considering the reaction temperatures. For instance, the dynamic toluene oxidation mechanism variation was triggered by the activation of the different active oxygen species[17,18]. In the low-temperature regime, gaseous oxygen molecules were directly converted to adsorbed oxygen species to facilitate the toluene oxidation[17,18]. With the increased temperature, the lattice oxygen from the bulk phase of $CeO_2$ supports gradually migrated to the interface and acted as the active surface lattice oxygen species to drive toluene oxidation[17,18]. A similar trend was detected for a dynamic transition of reaction mechanism from Langmuir-Hinshelwood to Mars-van Krevelen for toluene oxidation with rising temperature[19]. Furthermore, Li et al. found that the oxygen vacancy at the interfacial sites between Pt ensembles and $CeO_2$ was inactive in the low-temperature domain for the water-gas shift reaction. Once the temperature exceeded 180 °C, oxygen vacancy-$Ce^{3+}$ sites were stimulated and imitated to convert and supply active oxygen species to the interface[20]. Based on the above research, there might be a threshold temperature determining the oxygen activation to participate in the $C_3H_6$ oxidation reactions over $Pt_e/CeO_2$ catalysts. It is valuable to investigate the dynamic evolution properties to explicit the possible change in the reaction mechanism during the different temperature ranges. The results will benefit the understanding of the specific catalytic roles of $Pt_e$ catalysts in emission control applications.

Herein, Pt ensembles were loaded over the $CeO_2$ supports ($Pt_e/CeO_2$) via incipient wetness impregnation, which was further activated by $H_2$ reduction to improve the catalytic performance for $C_3H_6$ oxidation. High-angle annular darkfield scanning transmission electron microscopy (HAADF-STEM), extended X-ray adsorption fine structure (EXAFS), X-ray photoelectron spectroscopy (XPS), and catalytic performance tests unraveled that $H_2$ activation constructed metallic Pt ensembles locating at upper tiers of $CeO_2$ serving as the intrinsic active sites. In situ Raman spectra, near ambient pressure X-ray photoelectron spectroscopy (NAP-XPS), and Electron energy loss spectroscopy (EELS) results demonstrated that gaseous oxygen was activated at Pt-O-Ce interfacial sites to promote $C_3H_6$ oxidation above 170 °C, acting as the threshold temperature. In situ Diffuse Reflectance Infrared Fourier Transform Spectroscopy (DRIFTS), rigorous kinetic studies, and Density Functional Theory (DFT) calculations affirmed that the $C_3H_6$ coverage change and oxygen activation at the interfacial sites caused the dynamic transformation of the reaction models. The results will guide the precise design of $Pt_e$ catalysts and be also helpful for the understanding of their catalytic roles under varying reaction temperatures.

## Results

### Evaluation of the catalytic performance of $Pt_e/CeO_2$ catalysts

The study first tried to measure $C_3H_6$ oxidation light-off performance over $H_2$-activated $Pt_e/CeO_2$ catalysts, which significantly boosted the catalytic oxidation performance and could help explore the transformation of Pt ensemble structure and size during $H_2$ activation at different temperatures (Supplementary Fig. 1). As shown in Fig. 1a, $Pt_e$ barely reached 50% conversion of $C_3H_6$ ($T_{50}$) at approximately 282 °C, Meanwhile, $Pt_e$-300A ($Pt_e$ after $H_2$ reduction pretreatment at 300 °C) could significantly shift $T_{50}$ to low temperatures around 160 °C. According to Fig. 1b, $H_2$ activation caused a decline of the apparent activation energies from 138.0 to 111.5 kJ/mol for $Pt_e$ and $Pt_e$-300A, respectively. It was noteworthy that $Pt_e$-300A catalysts achieved an exceptional catalytic consumption rate for $C_3H_6$ oxidation compared to other Pt-based catalysts as shown in Supplementary Table 1, proving the considerable activity of activated $Pt_e$-300A catalysts (Fig. 1c). A similar promoting trend of $H_2$ activation was mirrored for the catalytic oxidation activities of $C_3H_6$ and/or CO, and the apparent activation energies of CO oxidation also dropped from 67.3 to 41.5 kJ/mol (Supplementary Figs. 2 and 3). Furthermore, the as-synthesized $Pt_e$-300A catalysts showed the same order of magnitude in CO consumption rate as the previously reported $Pt/CeO_2$ catalysts from Supplementary Table 2. The above results indicated that the $H_2$ activation could successfully fabricate efficient $Pt_e/CeO_2$ oxidation catalysts.

The synergistic interaction between Pt ensembles and $CeO_2$ was confirmed by comparing the catalytic performance between $Pt/CeO_2$ and $Pt/\gamma$-$Al_2O_3$ catalysts. To explore the interface effect of Pt metals and $CeO_2$ support, the catalytic performance of $Pt/\gamma$-$Al_2O_3$ with inert supports was also examined. Supplementary Fig. 4 exhibited that $H_2$-activated $Pt_e$-300A had better catalytic reactivity at a low-temperature regime than $Pt_e/\gamma$-$Al_2O_3$-300A catalysts. It indicated that the synergistic interaction between Pt and reducible $CeO_2$ supports was important in governing catalytic oxidation activity over $Pt_e$-300A catalysts. It is noteworthy that the physically mixed $Pt_e/\gamma$-$Al_2O_3$&$CeO_2$-300A catalysts obtained comparable activity toward $C_3H_6$ and CO oxidation to $Pt_e/\gamma$-$Al_2O_3$-300A catalyst. It implied that the proximity mattered for $Pt_e/CeO_2$ catalysts, and the synergistic interactions only occurred at the Pt-O-Ce interfacial sites. Moreover, $H_2$ activation had little effect on the catalytic oxidation activity of bare $CeO_2$ supports for $C_3H_6$ oxidation (Supplementary Fig. 5). The results revealed the transformation of Pt active sites after $H_2$ activation and synergistic interactions over Pt-O-Ce interfacial sites were two key factors in elevating oxidation performance of $Pt/CeO_2$ catalysts.

Water vapor is a common component as a key element influencing the performance and longevity of Pt-based emission control catalysts[21,22]. The co-feeding of 5% $H_2O$ did not reduce the light-off performance of $C_3H_6$ oxidation over $Pt_e$-300A (Supplementary Fig. 6). The catalytic activity remained stable in the presence of 5% $H_2O$, highlighting the promise of $Pt_e$-300A in practical applications for vehicle emission control (Supplementary Fig. 7). Meanwhile, $Pt_e$-300A catalysts was pretty stable without any change of Pt valence states during the stability tests, based on the XPS spectra of Pt $4f$ core-level analysis (Supplementary Fig. 8). No deactivation was observed for $Pt_e$-300A samples after cycling tests, also confirming the thermal stable properties of $H_2$-activated catalysts (Supplementary Fig. 9).

HAADF-STEM was carried out to analyze the geometry of Pt species on pristine and activated catalysts, which might be undergoing a dramatic reconstruction after $H_2$ activation. A few Pt single atoms and a single-layer Pt ensemble with a mean diameter of 0.45 nm could be detected on pristine $Pt_e$ catalysts (Fig. 2a and Supplementary Fig. 10a, b). $H_2$ activation caused a transformation of Pt species to form multilayer Pt ensembles, with a mean diameter of 0.84 nm (Fig. 2b and Supplementary Fig. 10c, d). In situ DRIFTS experiments of CO adsorption were further performed to investigate the chemical states

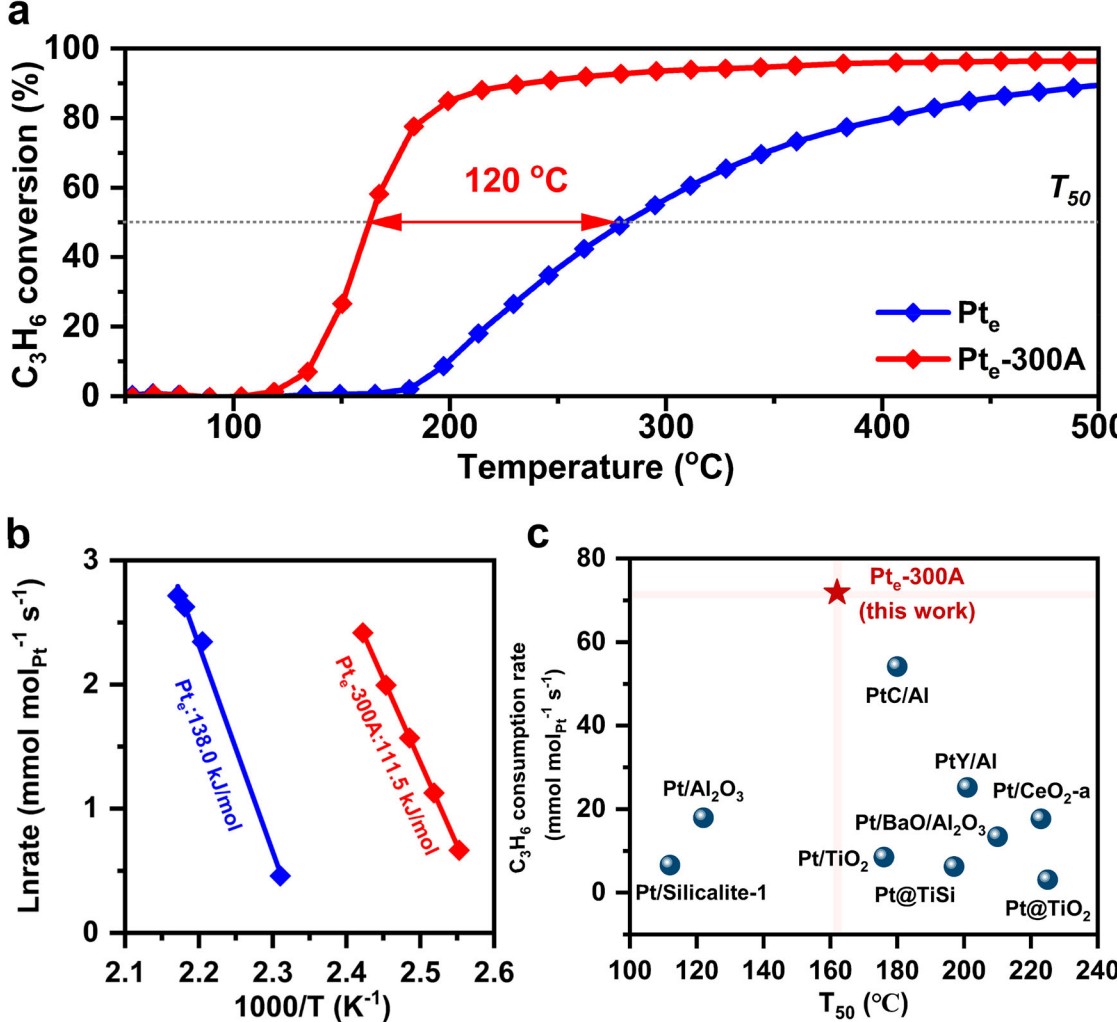

**Fig. 1 | Catalytic activities and apparent activation energies of C₃H₆ oxidation over supported Pt catalysts. a** $C_3H_6$ oxidation light-off curves. Reaction condition: 1000 ppm $C_3H_6$, and 10% $O_2$ in $N_2$ balance with a WHSV of 240,000 mL g$^{-1}$ h$^{-1}$; **b** Arrhenius plots of $C_3H_6$ oxidation; **c** Comparison of the reactivity of $C_3H_6$ oxidation between Pt$_e$-300A and previously reported Pt-based catalysts. For details for comparison, see Supplementary Table 1.

of surface Pt species. As shown in Fig. 2c, three major peaks at 2094, 2079, and 2036 cm$^{-1}$ were deconvoluted from the IR bands on Pt$_e$ catalysts, which could be ascribed to linearly bound CO adsorbed on Pt single atoms, the well-coordinated terrace sites, and the under-coordinated corner sites on Pt$^{\delta+}$ ensembles, respectively[23–25]. For Pt$_e$-300A catalysts with hydrogen activation, an intense band with two sub-peaks at approximately 2051 and 2033 cm$^{-1}$ appeared, corresponding to the linear CO adsorption at the well-coordinated terrace sites and under-coordinated edge sites within Pt$^0$ ensembles, respectively[26–28]. The extra peak at 1983 cm$^{-1}$ could be attributed to the bridging bound CO adsorbed on Pt ensembles[29], affirming an enlarged Pt ensembles after H$_2$ activation[30]. Moreover, the band at 2072 cm$^{-1}$ remained on Pt$_e$-300A as the CO linearly bound on unreduced oxidized Pt species. These newly created bands illustrated that the Pt ensembles over Pt$_e$-300A catalysts were composed of both Pt$^0$ and Pt$^{\delta+}$ sites.

The XAS data were further collected to elucidate the changes in the oxidation states and specific local coordination environments of Pt species after H$_2$ activation. Figure 2d compared the X-ray absorption near edge structure (XANES) spectra between Pt$_e$ and Pt$_e$-300A catalysts using Pt foil and PtO$_2$ as the reference samples. The edge position within Pt L$_3$-edge XANES data of Pt$_e$ was close to that of PtO$_2$, suggesting the single-layer Pt ensembles mainly existed in the highly oxidized states. Furthermore, the edge position shifted to lower

energy after H$_2$ activation, suggesting a more reduced Pt state over Pt$_e$-300A versus as-prepared Pt$_e$ samples. As displayed in Fig. 2e, the EXAFS spectra were plotted in R space to investigate the local structures. Pt$_e$ catalysts only demonstrated the first Pt-O coordination shell at 2.00 Å, implying that Pt ensembles were predominantly bound with the surface oxygen on CeO$_2$ (100) facets. The fitting model of the Pt$_e$-300A catalysts included both metallic Pt–Pt and Pt–O bonds, which indicated the formation of Pt$^0$ ensemble sites over the top layer of Pt$^{\delta+}$ planar (Supplementary Figs. 11 and 12). Meanwhile, the shorter Pt-Pt bond distance over Pt$_e$−300A (2.74 Å), compared to the Pt foil (2.76 Å), could be ascribed to the increment of the local electron density between two adjacent metal atoms triggered by the rehybridization of the *spd* orbitals in metal clusters[31,32]. This phenomenon usually occurs in the small metal nanoclusters[33], coinciding with the relatively small average diameter of the Pt ensembles over Pt$_e$−300A catalysts. Additionally, Fig. 2f showed the wavelet transform analysis based on Pt L$_3$-edge EXAFS oscillations, where the Pt$_e$-300A plots comprised Pt−O and Pt−Pt bonds. In contrast, only Pt-O bonds could be identified over Pt$_e$ catalysts, revealing that H$_2$ activation created metallic Pt ensembles as the major species, accompanied by partially oxidized Pt. Additional XRD and N$_2$ physisorption results proved that H$_2$ activation did not significantly affect the textural properties of CeO$_2$ supports (Supplementary Figs. 13, 14).

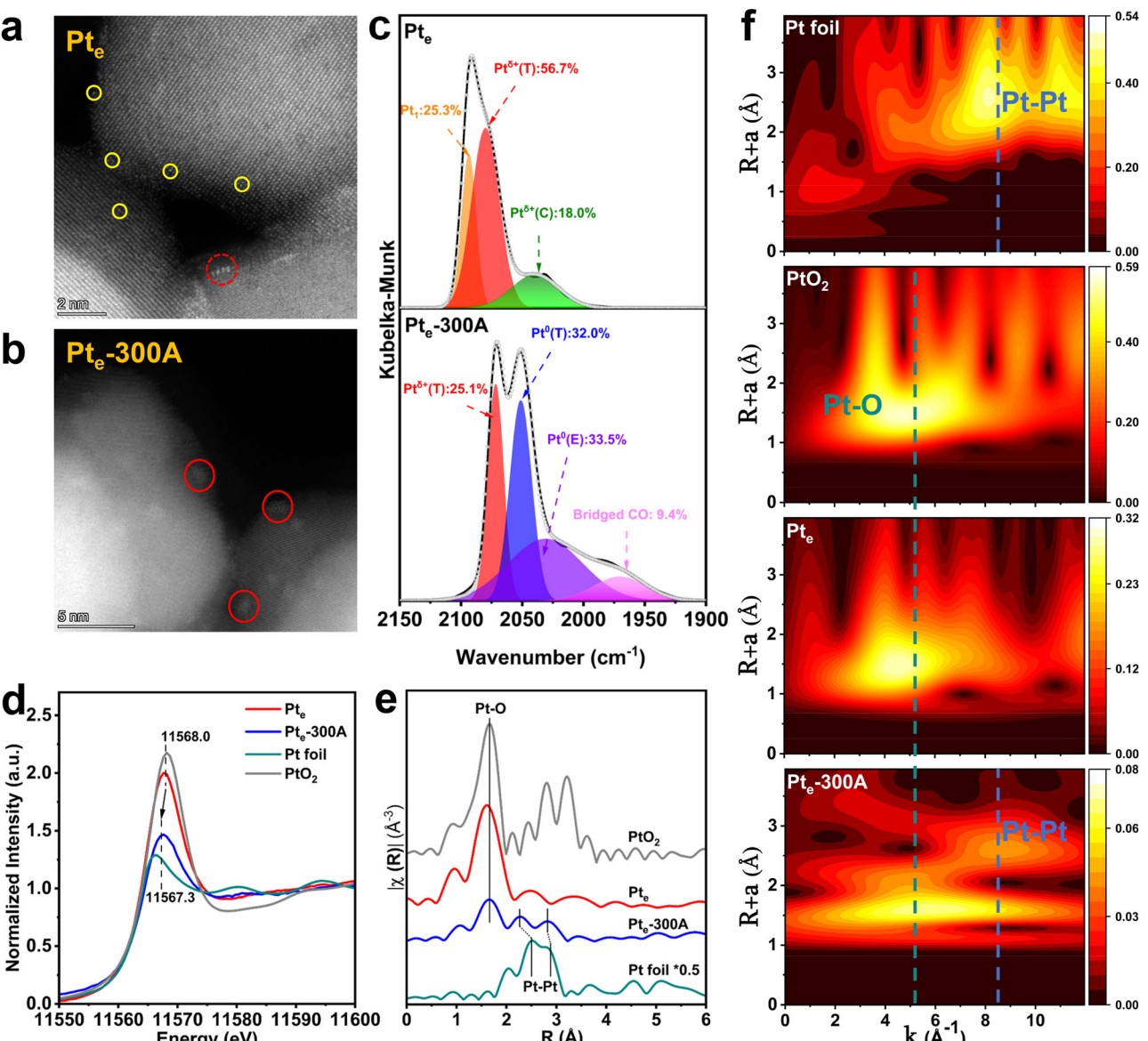

**Fig. 2 | Structural characterization and identification of intrinsic active sites over Pt$_e$ and Pt$_e$-300A catalysts.** HAADF-STEM images of **a** Pt$_e$ and **b** Pt$_e$-300A (yellow cycle: Pt single atoms; red dashed cycle: planner single-layer Pt ensembles; red solid cycle: multilayer Pt ensembles); **c** in situ DRIFTS spectra of CO adsorption at 30 °C on Pt$_e$ and Pt$_e$-300A; **d** normalized Pt L$_3$-edge XANES and **e** Fourier-transformed $k^2$-weighted EXAFS spectra in R space for Pt$_e$ and Pt$_e$-300A; **f** wavelet transform plot of Pt L$_3$-edge EXAFS spectra for Pt foil, PtO$_2$, Pt$_e$, and Pt$_e$-300A samples.

XPS experiments were performed to measure the chemical states of the supported Pt species. As shown in Supplementary Fig. 15 and Supplementary Table 3, only Pt$^{2+}$ and Pt$^{4+}$ were detected on Pt$_e$ samples, suggesting that highly oxidized Pt species exclusively survived on CeO$_2$. Metallic Pt was formed after H$_2$ activation, the deconvoluted doublets shifted to 71.8, 72.7, 75.1, and 76.0 eV, corresponding to Pt$^0$ or Pt$^{2+}$ in Pt $4f_{7/2}$ spectra and Pt $4f_{5/2}$ spectra, respectively[34,35]. H$_2$-TPR profiles (Supplementary Fig. 16) further demonstrated a significant declination of the relative area in the range of 100–250 °C on Pt$_e$-300A catalysts, indicating the disappearance of oxidized Pt species such as PtO$_x$[36], which agreed with the lowered coordination number of Pt–O bond between Pt$_e$ and Pt$_e$-300A (4.2 ± 0.2 vs. 1.5 ± 0.1) from Supplementary Table 4.

Since two different Pt sites, including Pt$^0$ and Pt$^{\delta+}$, have been detected on Pt ensembles, it is unavoidable to directly compare the reactivity on these sites regarding C$_3$H$_6$ oxidation quantitatively, which would further validate the rationality of metallic Pt ensembles as the

unique intrinsic active sites. On one hand, DFT calculations were first conducted to precisely analyze the free energy changes of the oxygen-facilitated dehydrogenation of the $sp^3$ hybrid C-H bonds, which was recognized as the rate-determining step (RDS) for catalytic oxidation of propylene in the following section. As shown in Supplementary Figs. 17 and 18, Pt$^0$ sites on the top layers obtained a much lower energy barrier (1.53 eV) than both Pt$^{\delta+}$ at the bottom sites (2.41 eV) and single-layer Pt ensembles on pre-activated Pt$_e$ catalysts (1.68 eV) for the dehydrogenation process, which led to the much better catalytic activity of C$_3$H$_6$ oxidation on metallic Pt sites formed during the H$_2$-trigged structural evolution. Additionally, it was consistent that the abstraction of the C-H bonds on the methyl group over metallic Pt ensemble sites exhibited a similar magnitude of activation energy barrier, compared to the identical process that took place on Pd/Cu$_{55}$ clusters (1.43 eV)[37] and Pt$_2$Sn/Pt(111) surface (1.63 eV) catalysts[38]. Supplementary Fig. 19 demonstrated density of state (DOS) results based on the $d$-orbital of Pt ensembles for different sites, where the $d$-band of

$Pt^0$ was centered at a higher energy (−1.98 eV) compared to $Pt^{\delta+}$ (−2.58 eV), indicating an increment of the adsorption reactivity accompanied with the facilitation of $C_3H_6$ adsorption on metallic Pt sites[39]. As illustrated in Supplementary Fig. 20, the Bader charges of the upper-tier $Pt^0$ and bottom-layer $Pt^{\delta+}$ over $CeO_2$ surface were calculated to be −0.13e and +0.30e, respectively. It revealed that metallic Pt ensemble sites could provide many more electrons for the $C_3H_6$ adsorbed molecules. Moreover, the difference in charge density of the oxygen-facilitated dehydrogenation step illustrated a more frequent electron transfer over $Pt^0$ than $Pt^{\delta+}$ sites, where the correlated results were affirmed by the charge density difference of 1.06e and 0.89e for $-CH_3$ activation with the assistance of oxygen over $Pt^0$ and $Pt^{\delta+}$ sites, respectively. Furthermore, DOS calculations for $C_3H_6$ adsorbed at $Pt^0$ and $Pt^{\delta+}$ sites revealed distinct electronic interactions shown in Supplementary Fig. 21. At the $Pt^0$ site, a significant orbital hybridization between C and Pt occurred within the energy range from −5 to −10 eV. The broad overlap across multiple energy levels indicated strong electron cloud interactions. The C-Pt orbital hybridization was also present over the $Pt^{\delta+}$ site, yet the hybridized peaks exhibited markedly reduced intensity and narrower energy distribution. Meanwhile, strong hybridization at the $Pt^0$ site shifted the $d$-band center to lower energies to −3.02 eV, in comparison to −2.07 eV at the $Pt^{\delta+}$ site. Collectively, these results demonstrated more stable $C_3H_6$ adsorption at top adsorption sites and stronger interfacial interactions, thereby facilitating subsequent $C_3H_6$ activation. On the other hand, the comparison of inherent catalytic oxidation activity between $Pt^0$ and $Pt^{\delta+}$ sites was also elucidated by FTIR tests, where CO was used as the titration gas to measure the reactivity of different Pt sites. The transient reactions were conducted between the saturated adsorbed CO and the flowing $O_2$ (Supplementary Fig. 22). Metallic Pt exhibited a rapid CO consumption rate over $Pt_e$-300A catalysts, confirming its role served as the sole intrinsic active site for catalytic oxidation, demonstrating significantly better reactivity than $Pt^{\delta+}$. The universality of the metallic Pt ensembles was further examined on $Pt_e$/γ-$Al_2O_3$ catalysts. Supplementary Figs. 23–25 additionally confirmed a positive correlation between the increased catalytic oxidation reactivity and the rising ratio of metallic Pt sites regarding $Pt_e$/γ-$Al_2O_3$ catalysts with $H_2$ activation, validating that metallic Pt functioned as the intrinsic active sites on Pt ensemble clusters supported by γ-$Al_2O_3$.

### Investigation of interfacial property by in situ characterization techniques

The dynamic change of the reaction model was studied using in situ characterization methods to uncover the mystery of $C_3H_6$ oxidation over $Pt_e$-300A at different temperatures. Firstly, in situ Raman spectra were measured using the identical reaction conditions as light-off tests. Figure 3a, b demonstrated two prominent bands at 458 and 588 $cm^{-1}$, symbolizing the $F_{2g}$ symmetry mode of the $CeO_2$ fluorite structure and the defect-induced mode, respectively[40,41]. Meanwhile, the band at 859 $cm^{-1}$ could be assigned to peroxide species ($O_2^{2-}$)[40], and the bands at 1059 and 1166 $cm^{-1}$ could be assigned to the in-plane bend of C−H species[19]. The intensity ratio of $I_D/I_{F2g}$ remained constant around 23.0% as the temperature rose from 100 to 160 °C, revealing that the concentration of oxygen vacancies was relatively stable. Once the reaction temperature was further elevated and exceeded 170 °C, the oxygen vacancy defects were gradually annihilated due to the adsorption and activation of adsorbed oxygen molecules. At the same time, the concentrations of peroxide species were also decreased above 170 °C, which was in line with their catalytic roles in $C_3H_6$ oxidation. Moreover, an equivalent phenomenon was detected over CO oxidation in Supplementary Fig. 26, where the $I_D/I_{F2g}$ ratio abruptly dropped above 170 °C, confirming a similar oxygen activation process stimulated and participated in CO oxidation. Secondly, NAP-XPS spectra of Ce$3d$ were displayed in Fig. 3c and Supplementary Fig. 27 to

trace the transformation in chemical valence during the heating process. From 100 to 160 °C, the fraction of $Ce^{3+}$ species was relatively stable at approximately 25%. When the reaction temperature increased above 170 °C, the $Ce^{3+}$ concentration gradually declined. It illustrated that the ratio of $Ce^{3+}/(Ce^{3+}+Ce^{4+})$ showed a decreased trend with ramping reaction temperatures, implying that the oxygen vacancy defects over Pt-O-Ce interfacial sites were gradually replenished during the reaction at high temperatures. Lastly, the $Ce^{3+}$ ratio at the interfacial sites after catalytic reactions below and above the threshold temperature of approximately 170 °C was probed by EELS to unambiguously verify the change of surface vacancies. As shown in Fig. 3d, e, the interfacial section of $Pt_e$-300A catalysts exhibited a more pronounced yellowish hue in the color bar after the $C_3H_6$ oxidation reaction at 162 °C compared to that at 188 °C, indicating a higher $Ce^{3+}$ ratio at the lower temperature. This observation suggests that oxygen vacancy defects were filled by activated oxygen during $C_3H_6$ oxidation with ramping temperature and cause a greater proportion of $Ce^{4+}$ formation. By combining in situ Raman spectra, NAP-XPS, and EELS results, it could be concluded that 170 °C was the threshold temperature for $C_3H_6$ oxidation on $Pt_e$-300A catalyst by different reaction pathways.

### Evaluation of $C_3H_6$ oxidation mechanism by varying reaction temperatures

In situ DRIFTS experiments of steady-state reactions at various temperatures were further conducted over $Pt_e$ and $Pt_e$-300A catalysts to elucidate different surface intermediates during $C_3H_6$ oxidation, and the detailed assignment of IR spectra was ascribed to Supplementary Table 5. As shown in Supplementary Fig. 28, the bands at 1660, 1274, and 1622 $cm^{-1}$ were observed at 30 °C on $Pt_e$, which could be attributed to C=C stretching, $CH_2$ deformation of gaseous $C_3H_6$, and C=C stretching of the adsorbed $C_3H_6$ molecules, respectively[42–44]. Once the temperature was increased to 150 °C, gaseous $C_3H_6$ completely vanished, suggesting $C_3H_6$ activation at high temperatures. The characteristic bands at 1240 and 1284 $cm^{-1}$ appeared when the reaction temperature reached 200 °C, corresponding to the generation of surface acrolein and acrylate[45,46]. These species would be further converted to acetate (1463 and 1405 $cm^{-1}$)[46]. As shown in Fig. 4a, b, the IR spectra of $Pt_e$-300A demonstrated complex adsorbed species at 30 °C. The bands at 1496 $cm^{-1}$ and 1435 $cm^{-1}$ were attributed to π-allylic intermediates, which were generated by the hydrogen abstraction from the weak methyl group with $sp^3$ hybridization[47,48]. Once the temperature reached 100 °C, gaseous $C_3H_6$ still presented at $ca.$ 1658 and 1265 $cm^{-1}$, while acrolein at 1268 $cm^{-1}$ was generated as the successive intermediates[46]. Acrylate exhibited intensive bands at approximately 1640 and 1288 $cm^{-1}$, while acetate could be observed at 1459 and 1395 $cm^{-1}$ above 200 °C[49]. The intensity of acetate was gradually increased with further ramping temperature, suggesting the facilitation of acetate generation at high temperatures. It should be noted that formate species might also be formed accompanying acetate generation, due to the destructive oxidation of $C_3H_6$ by breaking C=C bonds. Yet, no obvious formate species were detected in the IR spectra, probably due to the thermal decomposition above 200 °C. Moreover, compared to the situation on $Pt_e$ sample, many distinctive bands of acetate were observed at corresponding temperatures, while the characteristic peaks of $CO_2$ were only detected on the surface of $Pt_e$-300A. Therefore, it suggested that the metallic Pt ensemble sites on $H_2$-activated catalysts stimulated the oxidation process of acrylate to acetate, which could be recognized as a prior step in producing the final products of $CO_2$ and $H_2O$.

Rigorous kinetic studies were further conducted to evaluate the elementary reaction steps of $C_3H_6$ oxidation over $Pt_e$-300A catalysts. Figure 4c exhibited a sub-linear dependence between the $C_3H_6$ consumption rate on the partial pressures of $C_3H_6$ and $O_2$ at 162 °C, which revealed classic Langmuir-Hinshelwood models on $Pt_e$-300A catalysts.

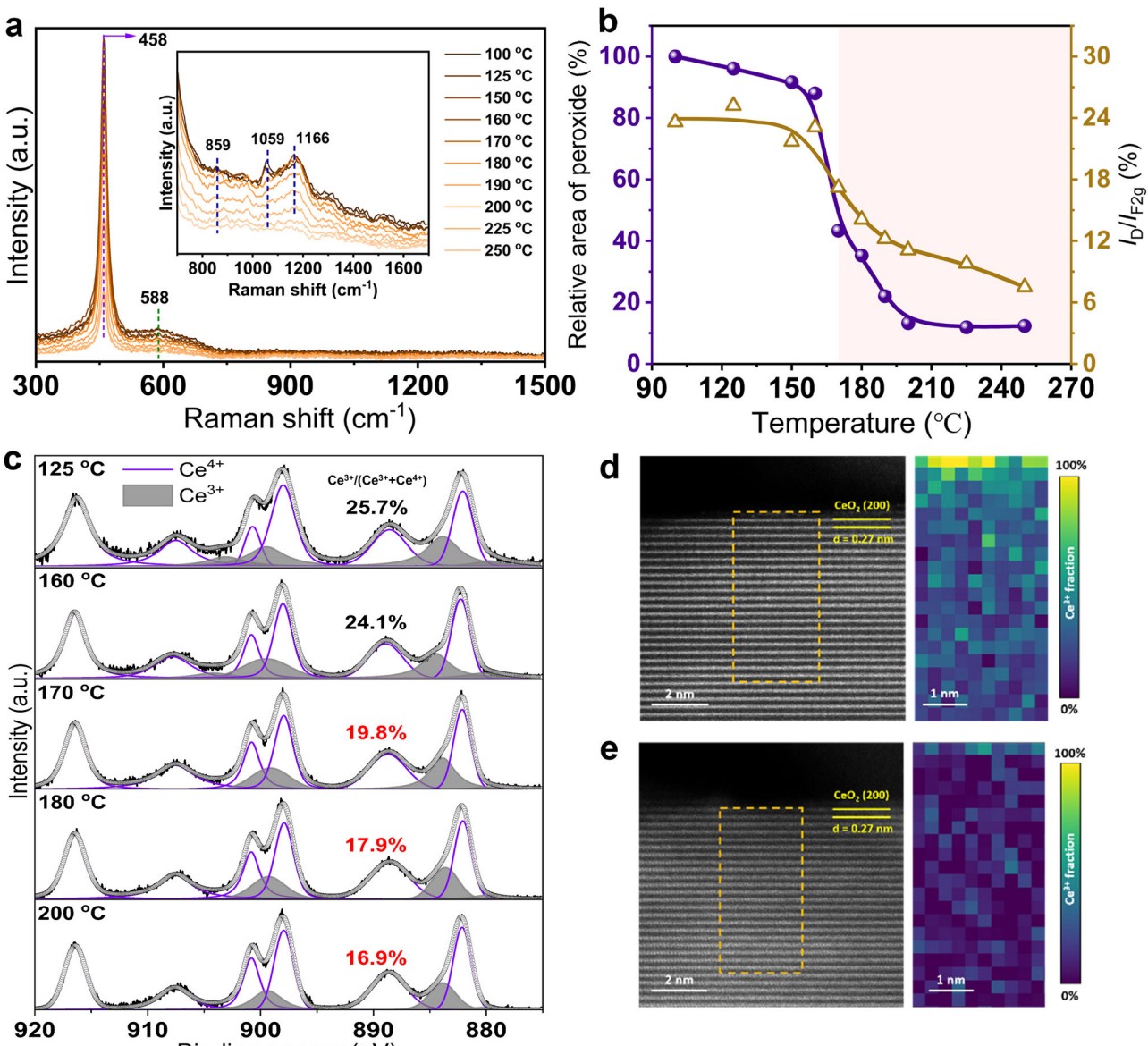

**Fig. 3 | Dynamic change of Pt$_e$-300A under C$_3$H$_6$ oxidation. a** In situ Raman spectra of C$_3$H$_6$ oxidation over Pt$_e$-300A from 100 to 250 °C; **b** Variation of peroxide ratio and I$_D$/I$_{F2g}$ ratio in the range of 100 to 225 °C; **c** NAP-XPS of Ce 3$d$ spectra at different temperatures; **d**, **e** EELS result of Pt$_e$-300A after catalytic reactions at 162 °C and 188 °C, respectively.

Based on the kinetics data at 162 °C, the elementary steps of C$_3$H$_6$ oxidation could be deduced and summarized in Supplementary Fig. 29. Initially, the gaseous O$_2$ was reversibly adsorbed on the vacancy sites (*) over metallic Pt ensemble sites (step 1), and then dissociated to produce the adsorbed O* atoms (step 2). Meanwhile, the quasi-equilibrium adsorption of C$_3$H$_6$ molecules occurred on the same sites and generated C$_3$H$_6$* (step 3). Subsequently, the dissociated O* kinetically activated the C-H bond by irreversibly coupling with the C$_3$H$_6$* to facilitate dehydrogenation and form surface-adsorbed C$_3$H$_5$* and OH* (step 4), which was recognized as RDS in the whole reaction process. The kinetics models of oxygen-facilitated dehydrogenation were similar to the catalytic oxidation of C$_3$H$_6$ over Ag/Al$_2$O$_3$ cluster catalysts[50]. The following process was induced by the quasi-equilibrated interaction between surface C$_3$H$_5$* and O*, generating adsorbed CO$_2$* and OH* (step 5). Ultimately, the final product of CO$_2$ was desorbed from the catalyst surface (step 6), and H$_2$O was generated from the reaction between OH* molecules and then desorbed by leaving the vacancy sites (steps 7 and 8). The derivation of

Supplementary Equation (1) was mainly based on the assumption of the pseudo-steady state for the kinetically observable O$_2$*, O*, and C$_3$H$_6$* species, accompanied by the quasi-equilibrium for steps 1–3 previous to the RDS. Supplementary Table 6 summarized the kinetic parameters derived by regressing the kinetics data to Supplementary Equation (1) while minimizing the residuals. Generally, C$_3$H$_6$ oxidation pathways started from the adsorbed and activation of C$_3$H$_6$ and O$_2$. Then, the reactions underwent C−H scission with O* to form an allyl intermediate (C$_3$H$_5$*), which was considered as kinetically-relevant step for the whole reaction.

DFT calculations were further conducted to explore the change of energies in the specific reaction route based on the elementary steps, and Supplementary Table 7 summarizes the potential energies in every reaction step. As shown in Fig. 4e, the reaction was initiated by the adsorption of O$_2$ on metallic Pt sites and then dissociated to the adsorbed O* molecules with a low dissociation energy of 0.10 eV (TS-1). The adsorbed O* was subsequently reacted with C$_3$H$_6$* adsorbed on metallic Pt ensembles with the activation energy of 1.53 eV (TS-2).

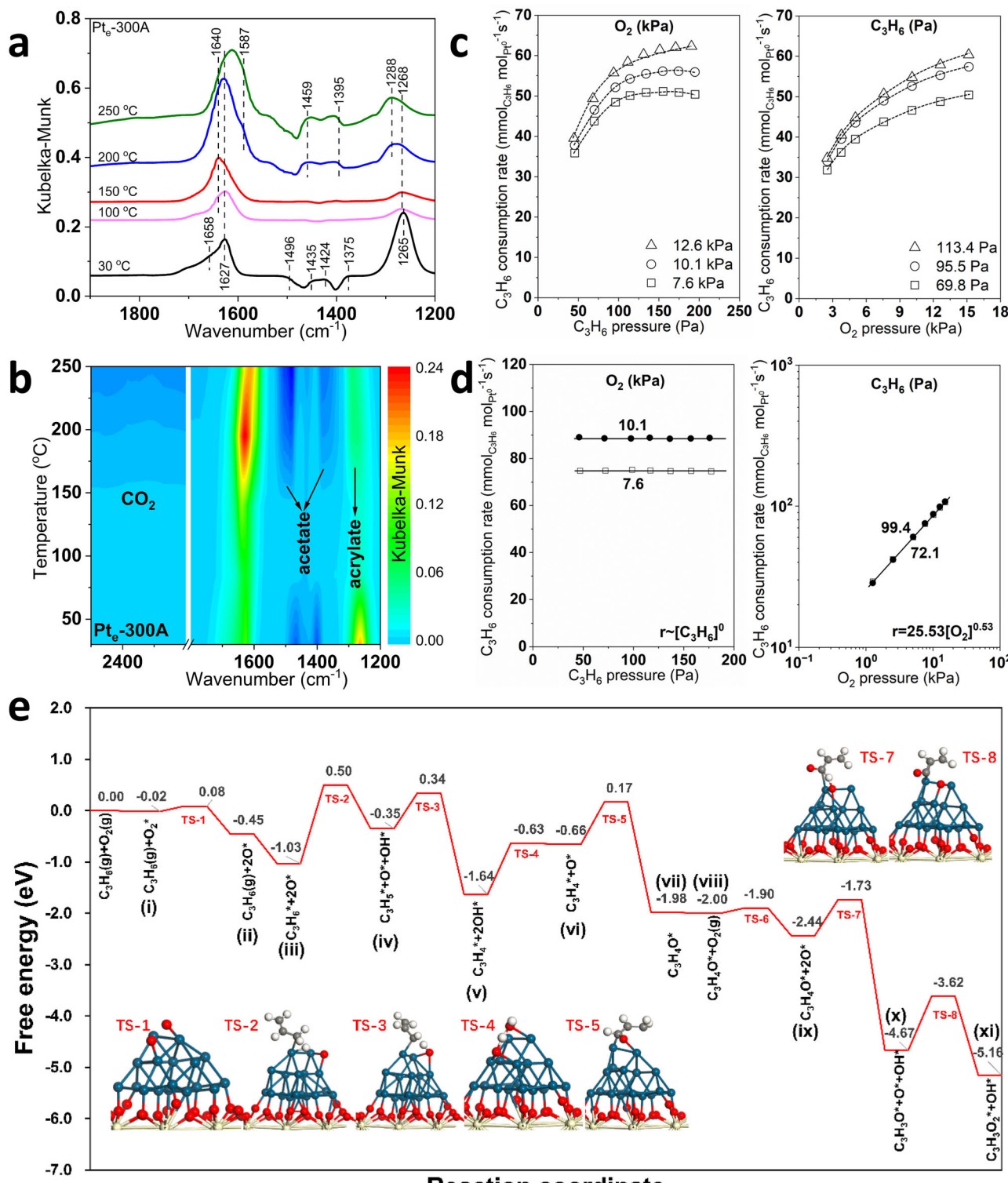

**Fig. 4 | Study of the reaction mechanisms and surface intermediates for $C_3H_6$ oxidation over $Pt_e$-300A. a** In situ DRIFTS spectra of steady-state $C_3H_6$ and $O_2$ co-adsorption; **b** Contour graphs for $C_3H_6$ oxidation; **c, d** Effects of $C_3H_6$ and $O_2$ partial pressures on $C_3H_6$ consumption rate over $Pt_e$-300A at 162 °C and 188 °C, respectively; **e** DFT calculations of $C_3H_6$ oxidation mechanisms and energy barriers with detailed transition states and free energy change.

Subsequently, the dehydrogenated $C_3H_5*$ was further activated by $O*$ to generate $C_3H_4*$ and $OH*$ with an energy barrier of 0.69 eV (TS-3). After the formed $OH*$ desorbed from the surface of metallic Pt ensembles with the energy of 1.01 eV (TS-4), the dissociated $O*$ would interact with $C_3H_4*$ to generate the surface-adsorbed acrolein ($C_3H_4O*$) with the reaction energy calculated as 0.83 eV (TS-5). Afterward, the second $O_2$ dissociation process occurred, with an identical activation energy to TS-1, to generate more $O*$ species (TS-6), which was then coupled with $C_3H_4O*$ to form $C_3H_3O*$ intermediates with the reaction energy of 0.71 eV (TS-7). Next, $C_3H_3O*$ acted as the crucial precursor of the carboxylates and was further coupled with the dissociated $O*$ to generate activated acrylate species ($C_3H_3O_2*$) with an activation energy

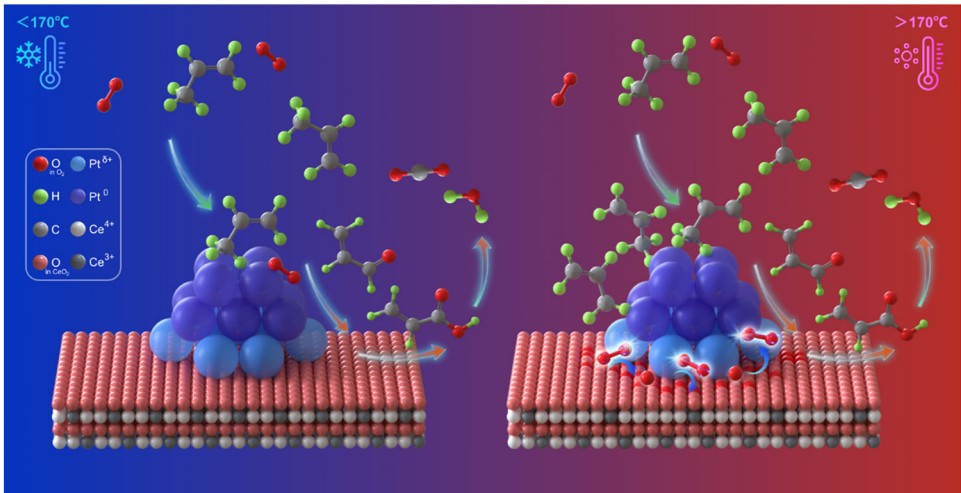

**Fig. 5 | The schematics of dynamic reaction pathways of $C_3H_6$ oxidation over Pt/CeO$_2$ ensemble catalysts.** At low temperatures (<170 °C), the reactions followed the classic Langmuir-Hinshelwood model, where both $C_3H_6$ and $O_2$ were adsorbed over metallic Pt ensembles. In contrast, at high temperatures (>170 °C), metallic Pt ensembles were fully covered by adsorbed $C_3H_6$, where $O_2$ was adsorbed and activated at vacant sites of Pt-O-Ce interfaces without competition with $C_3H_6$.

of 1.05 eV (TS-8). Lastly, $CO_2$* and $H_2O$* were produced and desorbed as the final products with a total free energy of −18.21 eV.

It was demonstrated that the reaction pathways of $C_3H_6$ oxidation followed classic Langmuir-Hinshelwood models based on the above kinetics and DFT results. Notably, the kinetic model of $C_3H_6$ oxidation might be changed by elevating the reaction temperature to 188 °C. Figure 4d revealed a zero-reaction order for $C_3H_6$ partial pressure, while a dependence with the half-reaction order of 0.53 was observed for $O_2$ partial pressure. The reaction order of approximately 0.5 regarding $O_2$ partial pressure was also observed for the kinetic studies of CO oxidation over Pt$_1$/CeO$_2$[51] and Rh/CeO$_2$[52] catalysts, which might prefer to occur on the supported catalysts with small metal clusters. It suggested that $C_3H_6$ molecules adsorbed stronger than $O_2$ and fully covered the surface of metallic Pt ensemble sites at 188 °C. Meanwhile, the adsorption and dissociation of $O_2$ took place at the vacant sites of the Pt-O-Ce interface (Pt-O$_v$-Ce). Supplementary Fig. 30 listed the sequence of fundamental steps consistent with the observed kinetics data of $C_3H_6$ oxidation once the reaction elevated to 188 °C. Compared to Supplementary Fig. 29, the major difference was $O_2$ adsorption and activation at the unoccupied oxygen vacancy (#) over Pt-O-Ce interfacial sites without competition with $C_3H_6$. Subsequently, the dissociated O# at the interfacial sites reacted with the adsorbed $C_3H_6$* at the Pt ensembles to finish the dehydrogenation of the $sp^3$ hybrid C-H bond and generate adsorbed $C_3H_5$* and OH#, which was irreversible and determined to be the RDS for the whole reaction. Supplementary Equation (2) could accurately describe the kinetics data of the reaction rates of $C_3H_6$ oxidation at 188 °C. Furthermore, the parity plots further verified the accuracy of the rate equation by comparing experimental and calculated data (Supplementary Fig. 31). The fractional coverages of $\theta(C_3H_6$*) could be further calculated based on the obtained kinetics data and related partial pressures of the reactants. It was found that the fractional coverages were higher than 0.99 for $\theta(C_3H_6$*) at temperatures of 188 °C, while the values would be only 0.20–0.55 for the same reaction at 162 °C. The results implied that the surface coverage of $C_3H_6$ over Pt$^0$ active sites was much more significant with increasing reaction temperatures, while oxygen would be adsorbed and activated at the Pt-O-Ce interfacial sites without competition with surface $C_3H_6$. Based on the above results, it was reasonable to deduce that there should be a threshold temperature affecting the reaction model during $C_3H_6$ oxidation. According to the findings, the dynamic reaction models based on different reaction temperature ranges were summarized and illustrated in Fig. 5.

It should be noted that the $C_3H_6$ oxidation might not follow the Mars-van Krevelen model at 188 °C in this study. On one hand, the Mars-van Krevelen model typically initiated the reaction between $C_3H_6$ and activation of lattice oxygen, followed by the activated oxygen (O*) filling into the oxygen vacancies. The reaction order of approximately 0.5 regarding $O_2$ partial pressure suggested the assumption that the elementary step of oxygen filling right after O* formation was the RDS. Yet, the fractions of Ce$^{3+}$ would gradually decrease with ramping reaction temperatures and caused an increase of Ce$^{4+}$ concentrations as evidenced by EELS tests and also in situ Raman and NAP-XPS study, suggesting that the oxidation of Ce$^{3+}$ was accelerated in the high temperature ranges. It meant that the oxygen activation and filling into vacancies was a fast reaction at high temperatures, and would not govern the whole catalytic reaction rate. On the other hand, theoretical results simulated that oxygen was adsorbed and activated at the top layer and interfacial Pt-O-Ce sites, which exhibited the energy barriers of 0.10 and 0.22 eV, respectively (Supplementary Table 8). Meanwhile, the energy barrier of dehydrogenation facilitated by interfacial oxygen was observed as 1.53 and 1.68 eV for the corresponding sites, respectively. It suggested that the oxygen-facilitated dehydrogenation was still the RDS step for $C_3H_6$ oxidation.

## Discussion

A highly efficient Pt/CeO$_2$ ensemble catalyst was obtained through a facile $H_2$ activation, which exhibited potential application toward vehicle emission control. HAADF-STEM and EXAFS results suggested the reconstruction from Pt single-layer planar (0.45 nm) to multilayer ensemble (0.84 nm) with the top layers composed of Pt$^0$ sites by hydrogen activation. In situ Raman spectra, NAP-XPS, and EELS experiments revealed that Ce$^{3+}$ defects and dioxygen intermediates were activated once the reaction temperature exceeded 170 °C. Based on the complementary experiments and theoretical results, Pt$^0$ ensemble sites demonstrated a low activation barrier for the dehydrogenation of $sp^3$ hybrid C-H with the assistance of oxygen, identified as the RDS, compared to Pt$^{δ+}$ sites. The kinetic results revealed that the $C_3H_6$ oxidation experienced a dynamic transformation of the reaction models, where the adsorption and dissociation of $O_2$ occurred at interfacial Pt-O$_v$-Ce sites after the temperature surpassed the threshold at approximately 170 °C. Combining the experimental phenomena, it was deduced that the involvement of interfacial Pt-O$_v$-Ce sites was the key to triggering the dynamic evolution of the reaction models. Generally, this research not only successfully clarified that the metallic Pt

ensembles were the intrinsic active sites for the highly efficient Pt/ CeO$_2$ catalysts regarding C$_3$H$_6$ oxidation, but also elucidated the dynamic change of the interfacial properties and the reaction model using in situ characterization methods. This work will guide the precise design and optimization of effective atomically dispersed platinum-based catalysts for emission control applications.

## Methods

### Catalyst preparation

**Materials.** All the chemicals are analytical reagent (AR) grades unless otherwise stated. Cerium nitrate hexahydrate (Ce(NO$_3$)$_3$·6H$_2$O) and sodium hydroxide (NaOH) powder were purchased from Sinopharm Chemical Reagent Co., Ltd. The noble metal precursor of tetra-ammineplatinum (II) nitrate (Pt(NH$_3$)$_4$(NO$_3$)$_2$) with 99.995% trace metals basis was obtained from Sigma-Aldrich. The γ-Al$_2$O$_3$ was bought from Sasol Chemical LLC. The deionized water was directly produced and collected via a Milli-Q ® water purification machine from Merck KGaA.

### Preparation of CeO$_2$ supports

A hydrothermal method was applied to synthesize supporting CeO$_2$ nanocubes[53]. 16.88 g NaOH and 1.96 g Ce(NO$_3$)$_3$·6H$_2$O were dissolved in 30 mL and 40 mL deionized water, respectively. After magnetic stirring for 15 min, the NaOH solution was added dropwise to Ce(NO$_3$)$_3$·6H$_2$O solution under vigorous stirring for another 30 min. Subsequently, the mixed solution was transferred to a 100 mL Teflon bottle, which was then tightly sealed in a stainless-steel vessel auto-clave and hydrothermally treated at 180 °C for 24 h. After cooling to room temperature, the white precipitate was washed and collected by centrifuging with deionized water at least three times and vacuum dried at 80 °C for 12 h. Finally, the yellowish products were calcined at 500 °C for 4 h with a ramping rate of 1 °C min$^{-1}$ to obtain CeO$_2$.

### Preparation of Pt$_e$/CeO$_2$ ensemble catalysts

0.5 wt.% Pt species were loaded on CeO$_2$ nanocubes via an incipient wetness impregnation method. 71 µL Pt(NH$_3$)$_4$(NO$_3$)$_2$ solution ([Pt] = 25 g L$^{-1}$) was applied as the Pt precursor and diluted in 0.9 mL deionized water. The aqueous solution was sonicated and slowly dripped onto 0.25 g ground CeO$_2$ powder. Subsequently, the mixture was evaporated at 60 °C for 1 h and oven-dried at 110 °C for 8 h. The resulting powder was calcined in static air at 500 °C for 6 h with a temperature ramp of 1 °C min$^{-1}$ to produce Pt ensemble nanoclusters denoting as Pt$_e$. For the hydrogen activation process, Pt$_e$ was pre-treated in 10% H$_2$/N$_2$ reducing flow under different temperatures for 1 h. The pretreated samples were denoted as Pt$_e$-XA, where X represented the activation temperature. Pt$_e$ catalysts were pretreated at 100, 200, 300, or 400 °C to identify the optimal activation condition. As evidenced in Supplementary Fig. 1, Pt$_e$-300A exhibited the highest catalytic activity towards C$_3$H$_6$ oxidation among the series of Pt$_e$-XA catalysts.

### Preparation of Pt$_e$/γ-Al$_2$O$_3$ ensemble catalysts

To conduct a comparison study, 0.5 wt% Pt$_e$/γ-Al$_2$O$_3$ catalysts were prepared by the identical procedure to Pt$_e$ catalysts. Prior to the synthesis, the commercial γ-Al$_2$O$_3$ was pretreated in a reducing flow of 10% H$_2$ with N$_2$ as the carrier gas at 350 °C for 2 h, leading to the formation of unsaturated Al$^{3+}_{penta}$ sites[54,55].

### Catalytic activity evaluation

**Light-off tests.** Reactant gases including 1% C$_3$H$_6$/N$_2$ (99.999%), 5% CO/ N$_2$ (99.999%) high-purified O$_2$ (99.999%), and high-purified N$_2$ (99.999%) were purchased from Shanghai Weichuang Standard Gas Analytical Technology Co., Ltd. The gas flow was precisely manipulated via mass flow controllers from Beijing Sevenstar Electronics Co., Ltd.

The catalytic light-off performance of C$_3$H$_6$ oxidation was measured in a fixed-bed quartz tube reactor with an internal diameter of 5.0 mm. Two thermocouples located upstream and downstream of the catalyst bed were utilized to monitor and regulate the temperature in the reactor. During each light-off test, the furnace reached the target temperature with a ramp rate of 2 °C min$^{-1}$. 50 mg granulated catalysts with mesh sizes of 180–250 µm were diluted by the same size quartz sands, which was pre-calcined at 800 °C for 6 h in static air before mixing, to minimize the heat effect. The total flow rate was maintained at 200 mL min$^{-1}$ for all reactions under atmospheric pressure, which corresponded to a weight hourly space velocity (WHSV) of 240,000 mL g$^{-1}$ h$^{-1}$. For C$_3$H$_6$ oxidation activity test, a feeding gas flow composed of 0.1% C$_3$H$_6$, 0.4% CO (when used), 5% H$_2$O (when used), 10% O$_2$ balanced with N$_2$ passed through the tube reactor. The reactant and product gas concentrations were collected and analyzed by a Fourier-transform infrared (IR) spectrometer (Antaris IGS Gas Analyzer, Thermo Fisher Scientific Inc.). A high-pressure syringe pump was applied to precisely manipulate the injection rate of deionized water, which was completely vaporized in a gasifier isothermal at 150 °C before pumping into the reactor system. The C$_3$H$_6$ and CO conversions ($X$) were calculated by the following equations:

$$X_{C_3H_6}(\%) = \frac{C_{C_3H_6,\,in} - C_{C_3H_6,\,out}}{C_{C_3H_6,\,in}} \times 100\% \tag{1}$$

$$X_{CO}(\%) = \frac{C_{CO,\,in} - C_{CO,\,out}}{C_{CO,\,in}} \times 100\% \tag{2}$$

where C$_{in}$ and C$_{out}$ represent the inlet and outlet concentrations of the reactants, respectively.

C$_3$H$_6$ and CO consumption rates (mmol mol$_{Pt}^{-1}$ s$^{-1}$) were collected at the constant temperature points and calculated by the following equation:

$$R = \frac{v \times M_{Pt}}{m_{Pt} \times V_m} \times X \tag{3}$$

Where $v$ represents the flow rate of the reactants (mL s$^{-1}$); $M_{Pt}$ is the atomic mass of platinum; $m_{Pt}$ is the mass of the platinum in the catalysts and determined by ICP-OES methods; $V_m$ is the molar volume of gas equals 24.5 L mol$^{-1}$ at room temperature.

### Apparent activation energy and kinetic evaluation

Arrhenius plots were constructed in a differential reactor by testing the C$_3$H$_6$ consumption rate at various temperature points following the identical reaction gas composition and WHSV to the light-off tests. All the catalytic conversions were strictly restricted below 15% to exclude the heat and mass transfer limitations. Meanwhile, the reactor was ramped to each target temperature at 2 °C min$^{-1}$ and held for at least 45 min to achieve a steady state. The apparent activation energy ($E_a$) was calculated by the Arrhenius plot.

For the kinetic experiments, the C$_3$H$_6$ oxidation rate was measured at a constant temperature by altering the partial pressure of the reactant gas without any back pressure in the reactor system. Heat and mass transfer limitations have been ruled out based on the theoretical results (Supplementary Table 9). 20 mg fine-ground catalysts with mesh sizes of 180–250 µm were mixed with 50 mg identical-size quartz sands, and packed into the fixed-bed tube reactor. Most catalytic conversions were restrained below 12%, maximizing at approximately 18%, which ensured that the kinetic evaluation could presume the differential reaction conditions. The mean partial pressure was determined as the average pressure between the inlet and outlet flows using

the following equations:

$$P_{C_3H_6} = \frac{P_{C_3H_6,\,in} + P_{C_3H_6,\,out}}{2} \qquad (4)$$

$$P_{O_2} = \frac{P_{O_2,\,in} + P_{O_2,\,out}}{2} \qquad (5)$$

The mean partial pressure of $C_3H_6$ and $O_2$ ranged from 45.52 to 195.04 Pa and 2.51 to 12.64 kPa, respectively.

## Inductively coupled plasma optical emission spectroscopy (ICP-OES)

The ICP-OES experiments were conducted on Avio 550 (PerkinElmer Inc.) to detect the actual Pt content of different catalysts.

## Scanning transmission electron microscopy (STEM)

The aberration-corrected scanning transmission electron microscopy was conducted on a Thermo Fisher Themis Z transmission electron microscope to analyze the morphology and elemental distribution of the catalysts. This instrument was operated at a working voltage of 300 kV and equipped with two aberration correctors. 4 in-column Super-X detectors were applied to conduct Energy-dispersive X-ray spectroscopy analysis. High angle annular dark-field (HAADF) STEM images were captured using a camera length of 115 mm on HAADF detectors with inner and outer collection angles of 47 and 200 mrad, respectively. EELS data were acquired using a Gatan Enfinium ER (model 977) EELS spectrometer with a dual EELS function.

## X-ray absorption spectroscopy (XAS)

The XAS scans were performed for $Pt_e$ and $Pt_e$-300A on 21 A X-ray nanodiffraction beamline (4-bounce channel-cut Si (111) monochromator) of Taiwan Photon Source (TPS) at the National Synchrotron Radiation Research Center (NSRRC). The measurements for the $Pt$-$L_3$ edge were carried out in the fluorescence mode in an energy range from 6 to 27 eV corresponding to the photon flux between $1 \times 10^{11}$ - $3 \times 10^9$ photon/s. The end-station is equipped with three ionization chambers and a Lytle/SDD detector after the focusing position of the KB mirror to collect data. The XANES data and EXAFS data were analyzed and fitted using the Athena and Artemis software from the Demeter software package, respectively.

## X-ray diffraction (XRD)

The XRD experiments were carried out on a LabX XRD-6100 instrument from Shimadzu Corporation, which operated at 40 mA and 40 kV with Cu Kα radiation ($\lambda = 0.15406$ nm) under an ambient condition. The scanning range of $2\theta$ angle was recorded from 5 to 90° in a step rate of 3.33° min$^{-1}$ to investigate the phase structure.

## N$_2$ adsorption and desorption isotherm

$N_2$ physisorption isotherm experiments were carried out on a Micromeritics ASAP 2020 analyzer. Prior to each measurement, the samples were firstly degassed under 300 °C for 8 h. The Barrett–Joyner–Halenda method was applied to determine the pore size distribution using the data of nitrogen desorption isotherm. The specific surface area was calculated using the Brunauer–Emmett–Teller equation.

## X-ray photoelectron spectroscopy (XPS)

The XPS spectra were collected on a Kratos AXIS Ultra DLD instrument (Shimadzu Corporation) operating at a working current and voltage of 8 mA and 14 kV, respectively, utilizing a monochromatic Al source. The binding energies of all tested elements were calibrated on the base of the standard C1s line at 284.8 eV.

## H$_2$ temperature-programmed reduction (H$_2$-TPR)

$H_2$-TPR experiments were performed on the AutoChem II 2920 chemisorption analyzer from Micromeritics to investigate the redox properties of the catalysts. 40 mg catalysts were loaded into the U-shape quartz tube, which was pretreated in a 50 mL min$^{-1}$ Ar flow at 300 °C for 30 min and then cooled down to room temperature. Once the baseline was stable at room temperature, the reactor was heated to 1000 °C at a ramp rate of 10 °C min$^{-1}$ under a reducing gas stream of 5% $H_2$/$N_2$ flow (50 mL min$^{-1}$). The signal was monitored and recorded with a thermal conductivity detector. For $Pt_e$-300A samples, an additional activation process was conducted after the Ar purge at 300 °C, and the catalyst was pretreated in a 5% $H_2$/Ar at the target temperature for 1 h prior to cooling to room temperature.

## In situ diffuse reflectance infrared Fourier transform spectroscopy (in situ DRIFTS)

In situ DRIFTS experiments were conducted on a Nicolet 6700 FTIR equipped with a mercury-cadmium-telluride detector cooling by liquid nitrogen. The fine-ground catalyst powder was loaded into a high-temperature reaction chamber with three $Ba_2F$ windows on the dome of the cell. The IR spectra resulted from averaging 64 scans at a resolution of 4 cm$^{-1}$.

$Pt_e$ and $Pt_e$-300A samples were pretreated in the $N_2$ flow (100 mL min$^{-1}$) and 10% $H_2$/$N_2$ (100 mL min$^{-1}$) at 300 °C for 1 h, respectively. Subsequently, the reaction chamber was cooled down to the target temperatures under $N_2$ gas flow. The background spectrum was recorded at each desired temperature and subtracted from the sample spectrum.

For the CO adsorption and $O_2$ purging experiments, 1% CO/$N_2$ in a total flowrate of 100 mL min$^{-1}$ was passed through the reaction chamber at the desired temperature. Once the CO adsorption saturated on the surface of catalysts, the system was purged by 100 mL min$^{-1}$ $N_2$ flow until the spectra became unchanged to remove the weakly adsorbed CO molecules. Finally, 10% $O_2$ was introduced into the reactor to investigate the reactivity of different active sites. DRIFTS spectra were recorded throughout the entire reaction process. For the $C_3H_6$ oxidation, a mixture gas flow composed of 0.4% $C_3H_6$ and 10% $O_2$ in $N_2$ balance (100 mL min$^{-1}$) was introduced into the reaction cell. The DRIFTS spectra were recorded at each temperature for at least 30 min.

## In situ Raman spectroscopy

A Horiba LabRam HR spectrometer using visible laser excitation with a wavelength of 514 nm emitted by a He-Cd laser was utilized for the in situ Raman experiments. A confocal microscope (Olympus BX-30-LWD) paired with a 50x long working distance objective was applied to focus the laser on the sample. The scattered photons were concentrated onto a single-stage monochromator and monitored using a UV-sensitive liquid nitrogen-cooled charge-coupled device (CCD) detector (Horiba CCD-3000 V). $Pt_e$-300A samples were pretreated in 10% $H_2$ at the tube reactor before the test. For $C_3H_6$ oxidation, the experiments were performed under the flowing reactant composed of 1000 ppm $C_3H_6$ and 10% $O_2$ balanced with $N_2$ at a total flowrate of 200 mL min$^{-1}$. For CO oxidation, the experiments were conducted under the flowing reactant composed of 4000 ppm CO and 10% $O_2$ balanced with $N_2$ at a total flowrate of 50 mL min$^{-1}$. Every spectrum was collected after the reaction for 20 min at each temperature point from 50 to 250 °C.

## NAP-XPS

The NAP-XPS were acquired using a SPECS-AU190069 instrument. The instrument features a multi-stage differential pumping system and a static voltage lens, making it suitable for usage in ultra-high vacuum ($1 \times 10^{-9}$ mbar) with gases ranging from 0 to 5 mbar. The spectra were obtained using monochromatized Al Kα irradiation (1486.6 eV),

generated by 50 W of excitation source power in an Al anode (SPECS XR-50). The X-ray spot was approximately 0.3 mm in diameter and located near the nozzle's opening. A pressure-reducing valve maintained a reaction pressure of 1 mbar. The powder sample was flattened into a smooth sheet and placed on a specially designed sample table that may be heated during the reaction. An electron flood cannon was used to correct for the charging of catalysts during tests.

## Computational method

The DFT calculations were performed by the Vienna Ab initio Simulation Package (VASP 5.4.1)[56] with the projector augmented wave method[57]. The exchange-functional is treated using the generalized gradient approximation with Perdew-Burke-Emzerhof[58] functional. The energy cutoff for the plane wave basis expansion was set to 450 eV. Partial occupancies of the Kohn−Sham orbitals were allowed using the Gaussian smearing method and a width of 0.2 eV. The Brillouin zone was sampled with the Monkhorst−Pack k-point of $2 \times 2 \times 1$ was applied for all the calculations for surface structures. The $CeO_2(100)$ support was modeled as a four-layer slab, with the top two layers fully relaxed and the bottom two layers constrained. The pre-activated surface was represented by a supported single-layer $Pt_7$ cluster, while the post-activated state was modeled by a multi-layer $Pt_{17}$ cluster. The self-consistent calculations apply a convergence energy threshold of $10^{-5}$ eV, and the force convergency was set to 0.05 eV/Å. The free energy corrections were calculated by the following equation:

$$\Delta G = \Delta E + \Delta G_{ZPE} + \Delta G_U - T\Delta S \qquad (6)$$

where $\Delta E$, $\Delta G_{ZPE}$, $\Delta G_U$, and $\Delta S$ refer to the DFT calculated energy change, the correction from zero-point energy, the correction from inner energy, and the correction from entropy[59]. The transition state was located via constrained optimization based on a process of varying the target reaction coordinate while relaxing all other degrees of freedom. The optimized structure was subsequently validated by a vibrational frequency analysis, confirming the presence of exactly only one imaginary frequency[60,61].

## Reporting summary

Further information on research design is available in the Nature Portfolio Reporting Summary linked to this article.

# Data availability

The data generated in this study are provided in the Supplementary Information/Source Data file. Data are available from the corresponding authors upon request. Source data are provided with this paper.

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

## Acknowledgements

L.M. acknowledges the financial support by the National Key Research and Development Program of China (2024YFB4105100), the National Natural Science Foundation of China (22176122), the Shanghai Pujiang Program (20PJ1407000), the Oceanic Interdisciplinary Program of Shanghai Jiao Tong University (SL2022ZD104), and the Zhejiang Key Laboratory of Low-carbon Control Technology for Industrial Pollution, Zhejiang University of Technology (2025DTZL01 and 2025ZY01076). S.D. was supported by the National Natural Science Foundation of China (22376062) and the Science and Technology Commission of Shanghai Municipality (24DX1400200 and 22ZR1415700). H.C. was supported by the National Natural Science Foundation of China (22176217) and the National Engineering Laboratory for Mobile Source Emission Control Technology (NELMS2018A12). F.L. thanks the Startup Fund from the University of California, Riverside (UCR).

## Author contributions

L.M. conceived and designed experiments. Z.-H.L. prepared samples, performed characterization, conducted catalytic performance tests; Y.L. and S.D. performed TEM tests; X.C., H.C., Z.-G.L. and K.Y. conducted research, discussed results and commented on the manuscript. Z.-H.L., F.L. and L.M. discussed results, wrote and revised the manuscript. N.Y. supervised the research. All the authors have given approval to the final version of the manuscript.

## Competing interests

The authors declare no competing interests.
