## [Transparent Peer Review file · Nature Communications]

Temperature-Driven Mechanistic Transition in Propylene Oxidation over Pt/CeO₂ Ensemble Catalysts

Corresponding Author: Professor Fudong Liu

Version 1:

Reviewer comments:

Reviewer #1

(Remarks to the Author)

This study investigated the temperature-dependent reaction mechanisms of propylene (C₃H₆) oxidation over Pt/CeO₂ ensemble catalysts. The results clarified that metallic Pt ensembles serve as intrinsic active sites for propylene oxidation, resolving long-standing debates on the active species in Pt/CeO₂ catalysts. The findings revealed a temperature-driven dynamic transition (~170 °C threshold) in reaction mechanisms, where oxygen activation shifted from metallic Pt ensembles to Pt-O-Ce interfaces, offering new insights into how metal-support interactions evolve with temperature. This challenges the static view of catalytic mechanisms and highlights the critical role of interfacial sites in high-temperature reactivity. The integration of in situ spectroscopy, kinetics, and DFT provides a robust framework for understanding dynamic oxygen activation, guiding the design of efficient, temperature-adaptive catalysts for vehicle emission control. These findings advance fundamental catalysis science and have direct implications for developing sustainable exhaust treatment technologies. The following comments could be considered for minor revision.

1. It is not clearly stated why the pretreatment temperature of 300 °C was chosen for the Pt/CeO₂ ensemble catalysts. Since pretreatment temperature can significantly influence the oxidation state, dispersion, and interaction between Pt and the CeO₂ support, a brief justification for this specific choice would improve clarity. If possible, including a comparative experiment or citing prior studies that validate the use of 300 °C would help strengthen the experimental rationale. This would ensure that the observed catalytic behavior is representative and not an artifact of the chosen pretreatment condition.
2. In the CO oxidation studies, catalytic activity was tested over both Pte and Pte-300A catalysts. However, the apparent activation energies were not reported. Providing these values would offer important kinetic insight into the performance differences between the two catalysts. In addition, it is suggested that CO oxidation be evaluated on the Pte/γ-Al₂O₃&CeO₂-300A catalyst to better assess the interplay between Pt ensembles and the CeO₂ component. Such data would provide stronger evidence for the proposed synergistic roles of Pt and CeO₂ in enhancing catalytic activity.
3. The concept of temperature-induced transitions in active sites, from metallic Pt to Pt-O-Ce interfacial sites, is a central and compelling aspect of this work. To further support the generality of this dynamic model, it would be helpful to conduct additional in situ characterization experiments under CO oxidation conditions. These studies could confirm whether a similar mechanistic transition occurs in other oxidation reactions, thereby reinforcing the broader applicability of the proposed model and demonstrating its relevance beyond propylene oxidation.
4. Supplementary Figure 4 lacks a legend or key explaining the meaning of the symbols used in the graph. Without this information, it is difficult for readers to interpret the data accurately. Please revise the figure or its caption to clearly define all symbols, lines, and markers used, ensuring that the figure is self-explanatory and consistent with the standards expected for supporting information.
5. The reported particle sizes in the text appear to be slightly inconsistent with the results presented in Supplementary Figure 21, which contains the TEM images and corresponding size distribution data. To maintain accuracy and consistency, it is recommended that the authors carefully cross-check these descriptions and ensure alignment between the written discussion and the supplementary figures. Any discrepancies should be clarified or corrected to avoid potential confusion.

Reviewer #2

(Remarks to the Author)

This manuscript presents a comprehensive study on a hydrogen-activated Pt/CeO₂ ensemble catalyst with enhanced performance for C₃H₆ oxidation. The authors demonstrate a structural transformation from single-layer Pt to multilayer Pt

ensembles upon H₂ activation and provide valuable mechanistic insights. The integration of advanced in situ characterization techniques and DFT calculations significantly reinforces the conclusions. However, several points require clarification or further elaboration to strengthen the manuscript:

1. Both metallic Pt⁰ and interfacial Pt–O–Ce sites are identified as important for C₃H₆ oxidation. Could the authors provide insights into the optimal ratio of these two active sites for maximizing catalytic performance? Additionally, is there any specific H₂-activation temperature or duration that leads to this optimal site distribution?
2. In Figure 4e, the DFT analysis focuses on the hydrogen-activated multilayer Pt structure. To make a convincing argument that Pt–O–Ce alone is insufficient for C₃H₆ oxidation, the authors should also include DFT calculations for the pre-activation structure, i.e., the single-layer planar Pt configuration. While Supplementary Figure 16 presents a comparison, the electronic structure of Pt–O–Ce in a multilayer ensemble could differ significantly from that in a single-layer configuration. A direct DFT comparison of the Pt structure before and after hydrogen-activation would offer a comprehensive understanding of how structural evolution influences catalytic activity.
3. In Supplementary Figure 17, the density of states (DOS) plots for Pt⁰ and Pt^{δ+} sites appear quite similar, despite the reported d-band centers at -1.98 eV and -2.58 eV, respectively. Given the relatively small difference, this discrepancy may not be substantial enough to support definitive conclusions. To enhance the analysis, the authors should present the DOS of the propylene molecule adsorbed on both Pt⁰ and Pt^{δ+} sites, with particular attention to the HOMO structure of the carbon atoms, to better illustrate the electron-donating behavior the carbon atoms on these two types of sites.
4. The manuscript lacks details regarding the transition state calculation in the “Methods” section.

Version 2:

Reviewer comments:

Reviewer #1

(Remarks to the Author)

The authors have addressed all my concerns and the paper can be published as it.

Reviewer #2

(Remarks to the Author)

The authors have addressed all comments and questions satisfactorily, and the manuscript is deemed suitable for publication in Nature Communications.

Open Access This Peer Review File is licensed under a Creative Commons Attribution 4.0 International License, which permits use, sharing, adaptation, distribution and reproduction in any medium or format, as long as you give appropriate credit to the original author(s) and the source, provide a link to the Creative Commons license, and indicate if changes were

made.

Point-to-Point Response to the Reviewers' Comments:

Reviewer #1 (Remarks to the Author):

This study investigated the temperature-dependent reaction mechanisms of propylene (C_3H_6) oxidation over Pt/CeO₂ ensemble catalysts. The results clarified that metallic Pt ensembles serve as intrinsic active sites for propylene oxidation, resolving long-standing debates on the active species in Pt/CeO₂ catalysts. The findings revealed a temperature-driven dynamic transition (~ 170 °C threshold) in reaction mechanisms, where oxygen activation shifted from metallic Pt ensembles to Pt-O-Ce interfaces, offering new insights into how metal-support interactions evolve with temperature. This challenges the static view of catalytic mechanisms and highlights the critical role of interfacial sites in high-temperature reactivity. The integration of in situ spectroscopy, kinetics, and DFT provides a robust framework for understanding dynamic oxygen activation, guiding the design of efficient, temperature-adaptive catalysts for vehicle emission control. These findings advance fundamental catalysis science and have direct implications for developing sustainable exhaust treatment technologies. The following comments could be considered for minor revision.

(1)-Catalyst preparation

It is not clearly stated why the pretreatment temperature of 300 °C was chosen for the Pt/CeO₂ ensemble catalysts. Since pretreatment temperature can significantly influence the oxidation state, dispersion, and interaction between Pt and the CeO₂ support, a brief justification for this specific choice would improve clarity. If possible, including a comparative experiment or citing prior studies that validate the use of 300 °C would help strengthen the experimental rationale. This would ensure that the observed catalytic behavior is representative and not an artifact of the chosen pretreatment condition.

Response: Thank you very much for your kind suggestion. The light-off curves of C_3H_6 oxidation over supported Pt_e catalysts, which were pretreated in 10% H₂/N₂ reducing flow under different temperatures for 1 h, have been added to the revised manuscript. As shown in the figure below (Supplementary Fig. 1), Pt_e-300A exhibited the best catalytic activity, achieving the lowest T_{50} value compared to the supported Pt_e pretreated at 100, 200, and 400 °C (denoted as Pt_e-100A, Pt_e-200A, Pt_e-400A, respectively). These experimental results affirmed that 300 °C was the optimal activation temperature for supported Pt_e catalysts, maximizing their activities for CO and C_3H_6 oxidation reactions.

Modification: Middle/Page 26: The pretreated samples were denoted as Pt_e-XA, where X represented the activation temperature. Pt_e catalysts were pretreated at 100, 200, 300, or 400 °C to identify the optimal activation condition. As evidenced in Supplementary Fig. 1, Pt_e-300A exhibited the highest catalytic activity towards C_3H_6 oxidation among the series of Pt_e-XA catalysts.

Supplementary Fig. 1 Catalytic activities of C₃H₆ oxidation over supported Pt catalysts pretreated under different temperatures. Reaction condition: 1000 ppm C₃H₆ and 10% O₂ in N₂ balance with a WHSV of 240, 000 mL g⁻¹ h⁻¹.

(2)-Results

In the CO oxidation studies, catalytic activity was tested over both Pt_e and Pt_e-300A catalysts. However, the apparent activation energies were not reported. Providing these values would offer important kinetic insight into the performance differences between the two catalysts.

Response: Thank you very much for your valuable suggestions. Comparison of the apparent activation energies for CO oxidation over Pt_e and Pt_e-300A catalysts has been added to the revised manuscript. As shown in the figure below (Supplementary Fig. 3b), the apparent activation energy decreased from 67.3 to 41.5 kJ/mol, which was consistent with the trend observed for C₃H₆ oxidation in Fig. 1b.

Modification: Middle/Page 6: A similar promoting trend of H₂ activation was mirrored for the catalytic oxidation activities of C₃H₆ and/or CO, and the apparent activation energies of CO oxidation also dropped from 67.3 to 41.5 kJ/mol (Supplementary Figs. 2 and 3).

Supplementary Fig. 3 Catalytic activities and apparent activation energies of CO oxidation over supported Pt catalysts. a CO oxidation light-off curves; **b** Arrhenius plots of CO oxidation. Reaction condition: 4000 ppm CO and 10% O₂ in N₂ balance with a WHSV of 240, 000 mL g⁻¹ h⁻¹.

In addition, it is suggested that CO oxidation be evaluated on the Pt_e/γ-Al₂O₃&CeO₂-300A catalyst to better assess the interplay between Pt ensembles and the CeO₂ component. Such data would provide stronger evidence for the proposed synergistic roles of Pt and CeO₂ in enhancing catalytic activity.

Response: Thank you very much for your kind comments. The direct comparison of catalytic CO oxidation light-off performances over Pt_e-300A, Pt_e/γ-Al₂O₃, Pt_e/γ-Al₂O₃-300A, and Pt_e/γ-Al₂O₃&CeO₂-300A samples has been added to the revised manuscript. As shown in Supplementary Fig. 4b below, the physically mixed Pt_e/γ-Al₂O₃&CeO₂-300A catalysts obtained comparable activity toward CO oxidation to Pt_e/γ-Al₂O₃-300A catalysts, which was identical to the trend detected for C₃H₆ oxidation. However, the oxidation activity of Pt_e-300A at the low-temperature regime still outperformed Pt_e/γ-Al₂O₃-300A and Pt_e/γ-Al₂O₃&CeO₂-300A catalysts. This observation confirmed that the Pt-O-Ce structure formed at the interface between Pt and CeO₂ through synergistic interactions served as the key factor in enhancing low-temperature catalytic oxidation activity.

Modification: Top/Page 7: It is noteworthy that the physically mixed Pt_e/γ-Al₂O₃&CeO₂-300A catalysts obtained comparable activity toward C₃H₆ and CO oxidation to Pt_e/γ-Al₂O₃-300A catalyst. It implied that the proximity mattered for Pt_e/CeO₂ catalysts, and the synergistic interactions only occurred at the Pt-O-Ce interfacial sites.

Supplementary Fig. 4 Catalytic activities of C₃H₆ and CO oxidation over Pt_e-300A and different Pt/ γ -Al₂O₃ catalysts. a C₃H₆ and b CO oxidation light-off curves. Reaction condition: 1000 ppm C₃H₆, 4000 ppm CO, and 10% O₂ in N₂ balance with a WHSV of 240, 000 mL g⁻¹ h⁻¹. Pt/ γ -Al₂O₃&CeO₂-300A represented the physically mixed Pt/ γ -Al₂O₃ and CeO₂, which was activated by H₂ at 300 °C.

The concept of temperature-induced transitions in active sites, from metallic Pt to Pt-O-Ce interfacial sites, is a central and compelling aspect of this work. To further support the generality of this dynamic model, it would be helpful to conduct additional *in situ* characterization experiments under CO oxidation conditions. These studies could confirm whether a similar mechanistic transition occurs in other oxidation reactions, thereby reinforcing the broader applicability of the proposed model and demonstrating its relevance beyond propylene oxidation.

Response: Thank you very much for your constructive suggestion. We agree that additional *in situ* characterization using CO oxidation as a probe reaction would provide definitive proof for the feasibility of the proposed dynamic mechanism while simultaneously establishing its broader applicability. Therefore, the *in situ* Raman spectra experiments were carried out over Pt_e-300A sample under the flowing reactant composed of 4000 ppm CO and 10% O₂ balanced with N₂ at a total flow rate of 50 mL min⁻¹. As shown in the figure below (Supplementary Fig. 26), the ratio of I_D/I_{F2g} maintained at approximately 17.2% above 170 °C. However, the I_D/I_{F2g} ratio remained near 17.5% below 170 °C but abruptly decreased to 11.1% at 175 °C, followed by a progressive decline to 7.3% when the reaction temperature elevated to 225 °C. This observation confirmed that oxygen vacancy concentration remained stable below 170 °C but became activated above this threshold, indicating that the CO oxidation over Pt_e-300A catalyst also underwent a dynamic mechanistic evolution.

Modification: Middle/Page 15: Moreover, an equivalent phenomenon was detected over CO oxidation in Supplementary Fig. 26, where the I_D/I_{F2g} ratio abruptly dropped above 170 °C, confirming a similar oxygen activation process stimulated and participated in CO oxidation.

Modification: Top/Page 31: For CO oxidation, the experiments were conducted under the flowing reactant composed of 4000 ppm CO and 10% O₂ balanced with N₂ at a total flowrate of 50 mL min⁻¹.

Supplementary Fig. 26 Operando Raman spectra of Pt_c-300A under CO oxidation from 50 to 225 °C.

(3)-Supporting information

Supplementary Figure 4 lacks a legend or key explaining the meaning of the symbols used in the graph. Without this information, it is difficult for readers to interpret the data accurately. Please revise the figure or its caption to clearly define all symbols, lines, and markers used, ensuring that the figure is self-explanatory and consistent with the standards expected for supporting information.

Response: Thank you very much for your comments and suggestion. We have revised the previous Supplementary Fig. 4 by adding the legend of CeO₂ and CeO₂-300A.

Modification: Supporting information/Page S6: The revised figure was presented below. As the catalytic activity comparison of Pt-XA samples has been added as Supplementary Fig. 1, the original Supplementary Fig. 4 has been renumbered as Supplementary Fig. 5.

Supplementary Fig. 5 Catalytic activities of C₃H₆ oxidation over bare CeO₂ in the presence and absence of H₂ activation at 300 °C. Reaction condition: 1000 ppm C₃H₆, and 10% O₂ in N₂ balance with a WHSV of 240, 000 mL g⁻¹ h⁻¹.

The reported particle sizes in the text appear to be slightly inconsistent with the results presented in Supplementary Figure 21, which contains the TEM images and corresponding size distribution data. To maintain accuracy and consistency, it is recommended that the authors carefully cross-check these descriptions and ensure alignment between the written discussion and the supplementary figures. Any discrepancies should be clarified or corrected to avoid potential confusion.

Response: Thank you for your thorough review and valuable suggestions. We sincerely apologize for the inconsistency in significant figures across decimal values. To prevent potential misinterpretation, we have revised Fig. 21b to ensure precise alignment of diameter measurements with corresponding data in the supporting documentation. Additionally, Supplementary Fig. 21 has been renumbered as Supplementary Fig. 24 and is presented below.

Modification: Supporting information/Page S25: The morphology of Pt species over Pt_e/γ-Al₂O₃ and Pt_e/γ-Al₂O₃-300A catalysts was also captured and demonstrated. For Pt_e/γ-Al₂O₃ sample, Pt ensembles existed with a mean diameter of approximately 0.79 nm. Meanwhile, abundant Pt single atoms, with approximately 46%, could also be detected at the surface of Pt_e/γ-Al₂O₃ catalyst. The number of Pt single atoms was reduced dramatically after H₂ activation, accompanied by a significant rise in the average diameter of Pt ensembles (0.92 nm), which might be triggered by the migration of the Pt single atoms to the neighboring ensembles.

Supplementary Fig. 24 HAADF-STEM images and size distribution of Pt clusters over Pt_e/γ-Al₂O₃ and Pt_e/γ-Al₂O₃-300A catalysts. a, c Additional HAADF-STEM images of Pt_e/γ-Al₂O₃ and Pt_e/γ-Al₂O₃-300A catalysts, respectively (red cycle: Pt nanocluster; green cycle: Pt single atom); **b, d** size distribution of Pt ensembles over Pt_e/γ-Al₂O₃ and Pt_e/γ-Al₂O₃-300A catalysts, respectively.

Reviewer #2 (Remarks to the Author):

This manuscript presents a comprehensive study on a hydrogen-activated Pt/CeO₂ ensemble catalyst with enhanced performance for C₃H₆ oxidation. The authors demonstrate a structural transformation from single-layer Pt to multilayer Pt ensembles upon H₂ activation and provide valuable mechanistic insights. The integration of advanced in situ characterization techniques and DFT calculations significantly reinforces the conclusions. However, several points require clarification or further elaboration to strengthen the manuscript:

Both metallic Pt⁰ and interfacial Pt–O–Ce sites are identified as important for C₃H₆ oxidation. Could the authors provide insights into the optimal ratio of these two active sites for maximizing catalytic performance?

Response: Thank you for your valuable suggestions. We have conducted catalytic oxidation of propylene over Pt_e and Pt_e-XA catalysts, which were activated by H₂ at different temperatures (Supplementary Fig. 1). The series of catalysts were further characterized by XPS to understand the Pt valence ratio (Supplementary Fig. 15). As presented in Supplementary Fig. 15a, the Pt⁰ content in Pt_e-100A, Pt_e-200A, Pt_e-300A, and Pt_e-400A was quantified as 2.3%, 44.5%, 61.0%, and 75.6%, respectively. Given that Pt_e-300A demonstrated the best low-temperature catalytic activity for C₃H₆ oxidation (Supplementary Fig. 1), the optimal ratio between Pt⁰ and Pt-O-Ce (Pt⁰/Pt^{δ+}) was expected to be 1.6.

Modification: Supporting information/Page S16: Meanwhile, since Pt_e-300A demonstrated the best low-temperature catalytic activity for both C₃H₆ and CO oxidation among Pt_e-XA samples (Supplementary Fig. 1), the optimal ratio between Pt⁰ and Pt-O-Ce was expected to be 1.6.

Supplementary Fig. 15 XPS for Pt_e-XA and Pt_e catalysts. a Pt 4f, b O 1s, and c Ce 3d regions.

Additionally, is there any specific H₂-activation temperature or duration that leads to this optimal site distribution?

Response: Thank you very much for your kind suggestion. The light-off curves of C₃H₆ oxidation over supported Pt_e catalysts, which were pretreated in 10% H₂/N₂ reducing flow under different temperatures for 1 h, have been added to the revised manuscript. As shown in the figure below (Supplementary Fig. 1), Pt_e-300A exhibited the best catalytic activities, achieving the lowest T₅₀

values compared to the supported Pt_e pretreated at 100, 200, and 400 °C (denoted as Pt_e-100A, Pt_e-200A, Pt_e-400A, respectively). These experimental results affirmed that 300 °C was the optimal activation temperature for supported Pt_e catalysts, maximizing their activities for CO and C₃H₆ oxidation reactions.

Modification: Middle/Page 26: The pretreated samples were denoted as Pt_e-XA, where X represented the activation temperature. Pt_e catalysts were pretreated at 100, 200, 300, or 400 °C to identify the optimal activation condition. As evidenced in Supplementary Fig. 1, Pt_e-300A exhibited the highest catalytic activity towards C₃H₆ oxidation among the series of Pt_e-XA catalysts.

Supplementary Fig. 1 Catalytic activities of C₃H₆ oxidation over supported Pt catalysts pretreated under different temperatures. Reaction condition: 1000 ppm C₃H₆ and 10% O₂ in N₂ balance with a WHSV of 240, 000 mL g⁻¹ h⁻¹.

In Figure 4e, the DFT analysis focuses on the hydrogen-activated multilayer Pt structure. To make a convincing argument that Pt–O–Ce alone is insufficient for C₃H₆ oxidation, the authors should also include DFT calculations for the pre-activation structure, i.e., the single-layer planar Pt configuration. While Supplementary Figure 16 presents a comparison, the electronic structure of Pt–O–Ce in a multilayer ensemble could differ significantly from that in a single-layer configuration. A direct DFT comparison of the Pt structure before and after hydrogen-activation would offer a comprehensive understanding of how structural evolution influences catalytic activity.

Response: Thank you very much for your valuable comments and suggestions. DFT calculations probing methyl group dehydrogenation in C₃H₆ over pristine Pt_e catalysts with single-layer Pt ensembles have been conducted, and the results have been summarized into Supplementary Fig. 18. As shown in the figure below (Supplementary Figs. 17 and 18), Pt⁰ sites on activated Pt_e-300A exhibited a lower activation energy barrier (1.53 eV) versus single-layer Pt ensembles on pre-activated Pt_e catalysts (1.68 eV) for the dehydrogenation process. This confirmed that metallic Pt

sites formed during H₂-triggered structural evolution were key to enhancing low-temperature C₃H₆ oxidation activity.

Modification: Bottom/Page 11: As shown in Supplementary Figs. 17-18, Pt⁰ sites on the top layers obtained a much lower energy barrier (1.53 eV) than both Pt^{δ+} at the bottom sites (2.41 eV) and single-layer Pt ensembles on pre-activated Pt_e catalysts (1.68 eV) for the dehydrogenation process, which led to the much better catalytic activity of C₃H₆ oxidation on metallic Pt sites formed during the H₂-triggered structural evolution.

Supplementary Fig. 18 DFT calculations of dehydrogenation of methyl group in C₃H₆ at the as-synthesized, supported Pt catalysts. **a** DFT calculated optimized structures of the initial state (IS), transition state (TS), final state (FS) (Grey: carbon; White: hydrogen; Blue: Pt atoms; Red: oxygen; Pale yellow: cerium); **b** Energy barriers of oxygen-facilitated dehydrogenation on single-layer Pt ensemble.

In Supplementary Figure 17, the density of states (DOS) plots for Pt⁰ and Pt^{δ+} sites appear quite similar, despite the reported d-band centers at -1.98 eV and -2.58 eV, respectively. Given the relatively small difference, this discrepancy may not be substantial enough to support definitive conclusions. To enhance the analysis, the authors should present the DOS of the propylene molecule adsorbed on both Pt⁰ and Pt^{δ+} sites, with particular attention to the HOMO structure of the carbon atoms, to better illustrate the electron-donating behavior the carbon atoms on these two types of sites.

Response: Thank you very much for your constructive suggestions. We have conducted DOS calculations regarding C₃H₆ adsorption on both Pt⁰ and Pt^{δ+} sites over Pt_e-300A catalyst. As shown in the figure below (Supplementary Fig. 21), a notable orbital hybridization between C and Pt occurred over the top site (Pt⁰) within the -5 to -10 eV energy range, with pronounced C-Pt hybridized peaks. The broad overlap across multiple energy levels indicated strong electron cloud interactions. Even though C-Pt hybridization also occurred at the side site (Pt^{δ+}), the hybridized peaks exhibited markedly reduced intensity and narrower energy distribution compared to the top configuration. Meanwhile, strong hybridization at the Pt⁰ site shifted the d-band center to lower energies to -3.02 eV, in comparison to -2.07 eV at the Pt^{δ+} site. These results demonstrated stronger interfacial interactions and more stable chemisorption at Pt⁰ sites, thereby facilitating subsequent C₃H₆ activation.

Modification: Middle/Page 12: Furthermore, DOS calculations for C₃H₆ adsorbed at Pt⁰ and Pt^{δ+} sites revealed distinct electronic interactions shown in Supplementary Fig. 21. At the Pt⁰ site, a

significant orbital hybridization between C and Pt occurred within the energy range from -5 to -10 eV. The broad overlap across multiple energy levels indicated strong electron cloud interactions. The C-Pt orbital hybridization was also present over the Pt^{δ+} site, yet the hybridized peaks exhibited markedly reduced intensity and narrower energy distribution. Meanwhile, strong hybridization at the Pt⁰ site shifted the d-band center to lower energies to -3.02 eV, in comparison to -2.07 eV at the Pt^{δ+} site. Collectively, these results demonstrated more stable C₃H₆ adsorption at top adsorption sites and stronger interfacial interactions, thereby facilitating subsequent C₃H₆ activation.

Supplementary Fig. 21 Density of states projected on the 5d-orbital of Pt and 2p-orbital of C for C₃H₆ adsorption at a Pt⁰ and b Pt^{δ+} sites. The indicated number represents the d-band center for 5d-orbital of Pt.

The manuscript lacks details regarding the transition state calculation in the “Methods” section.

Response: Thank you for your valuable comment. We have added the details of the transition state calculation in the “Methods” section. We also supplemented much more information for Pt/CeO₂ ensemble catalyst models in this section.

Modification: Top/Page 32: The transition state was located via constrained optimization based on a process of varying the target reaction coordinate while relaxing all other degrees of freedom. The optimized structure was subsequently validated by a vibrational frequency analysis confirming the presence of exactly only one imaginary frequency^{60,61}.

Reference

60 Liu, Z. P. & Hu, P. General rules for predicting where a catalytic reaction should occur on metal surfaces: a density functional theory study of C–H and C–O bond breaking/making on flat, stepped, and kinked metal surfaces, *J. Am. Chem. Soc.* **125**, 1958–1967, (2003).

61 Alavi, A., Hu, P., Deutsch, T., Silvestrelli, P.L. & Hutter, J. CO oxidation on Pt (111): an
a
b

Point-to-Point Response to the Reviewers' Comments:

Reviewer #1 (Remarks to the Author):

This study investigated the temperature-dependent reaction mechanisms of propylene (C_3H_6) oxidation over Pt/CeO₂ ensemble catalysts. The results clarified that metallic Pt ensembles serve as intrinsic active sites for propylene oxidation, resolving long-standing debates on the active species in Pt/CeO₂ catalysts. The findings revealed a temperature-driven dynamic transition (~ 170 °C threshold) in reaction mechanisms, where oxygen activation shifted from metallic Pt ensembles to Pt-O-Ce interfaces, offering new insights into how metal-support interactions evolve with temperature. This challenges the static view of catalytic mechanisms and highlights the critical role of interfacial sites in high-temperature reactivity. The integration of in situ spectroscopy, kinetics, and DFT provides a robust framework for understanding dynamic oxygen activation, guiding the design of efficient, temperature-adaptive catalysts for vehicle emission control. These findings advance fundamental catalysis science and have direct implications for developing sustainable exhaust treatment technologies. The following comments could be considered for minor revision.

(1)-Catalyst preparation

It is not clearly stated why the pretreatment temperature of 300 °C was chosen for the Pt/CeO₂ ensemble catalysts. Since pretreatment temperature can significantly influence the oxidation state, dispersion, and interaction between Pt and the CeO₂ support, a brief justification for this specific choice would improve clarity. If possible, including a comparative experiment or citing prior studies that validate the use of 300 °C would help strengthen the experimental rationale. This would ensure that the observed catalytic behavior is representative and not an artifact of the chosen pretreatment condition.

Response: Thank you very much for your kind suggestion. The light-off curves of C_3H_6 oxidation over supported Pt_e catalysts, which were pretreated in 10% H₂/N₂ reducing flow under different temperatures for 1 h, have been added to the revised manuscript. As shown in the figure below (Supplementary Fig. 1), Pt_e-300A exhibited the best catalytic activity, achieving the lowest T_{50} value compared to the supported Pt_e pretreated at 100, 200, and 400 °C (denoted as Pt_e-100A, Pt_e-200A, Pt_e-400A, respectively). These experimental results affirmed that 300 °C was the optimal activation temperature for supported Pt_e catalysts, maximizing their activities for CO and C_3H_6 oxidation reactions.

Modification: Middle/Page 26: The pretreated samples were denoted as Pt_e-XA, where X represented the activation temperature. Pt_e catalysts were pretreated at 100, 200, 300, or 400 °C to identify the optimal activation condition. As evidenced in Supplementary Fig. 1, Pt_e-300A exhibited the highest catalytic activity towards C_3H_6 oxidation among the series of Pt_e-XA catalysts.

Supplementary Fig. 1 Catalytic activities of C₃H₆ oxidation over supported Pt catalysts pretreated under different temperatures. Reaction condition: 1000 ppm C₃H₆ and 10% O₂ in N₂ balance with a WHSV of 240, 000 mL g⁻¹ h⁻¹.

(2)-Results

In the CO oxidation studies, catalytic activity was tested over both Pt_e and Pt_e-300A catalysts. However, the apparent activation energies were not reported. Providing these values would offer important kinetic insight into the performance differences between the two catalysts.

Response: Thank you very much for your valuable suggestions. Comparison of the apparent activation energies for CO oxidation over Pt_e and Pt_e-300A catalysts has been added to the revised manuscript. As shown in the figure below (Supplementary Fig. 3b), the apparent activation energy decreased from 67.3 to 41.5 kJ/mol, which was consistent with the trend observed for C₃H₆ oxidation in Fig. 1b.

Modification: Middle/Page 6: A similar promoting trend of H₂ activation was mirrored for the catalytic oxidation activities of C₃H₆ and/or CO, and the apparent activation energies of CO oxidation also dropped from 67.3 to 41.5 kJ/mol (Supplementary Figs. 2 and 3).

Supplementary Fig. 3 Catalytic activities and apparent activation energies of CO oxidation over supported Pt catalysts. a CO oxidation light-off curves; **b** Arrhenius plots of CO oxidation. Reaction condition: 4000 ppm CO and 10% O₂ in N₂ balance with a WHSV of 240, 000 mL g⁻¹ h⁻¹.

In addition, it is suggested that CO oxidation be evaluated on the Pt_e/γ-Al₂O₃&CeO₂-300A catalyst to better assess the interplay between Pt ensembles and the CeO₂ component. Such data would provide stronger evidence for the proposed synergistic roles of Pt and CeO₂ in enhancing catalytic activity.

Response: Thank you very much for your kind comments. The direct comparison of catalytic CO oxidation light-off performances over Pt_e-300A, Pt_e/γ-Al₂O₃, Pt_e/γ-Al₂O₃-300A, and Pt_e/γ-Al₂O₃&CeO₂-300A samples has been added to the revised manuscript. As shown in Supplementary Fig. 4b below, the physically mixed Pt_e/γ-Al₂O₃&CeO₂-300A catalysts obtained comparable activity toward CO oxidation to Pt_e/γ-Al₂O₃-300A catalysts, which was identical to the trend detected for C₃H₆ oxidation. However, the oxidation activity of Pt_e-300A at the low-temperature regime still outperformed Pt_e/γ-Al₂O₃-300A and Pt_e/γ-Al₂O₃&CeO₂-300A catalysts. This observation confirmed that the Pt-O-Ce structure formed at the interface between Pt and CeO₂ through synergistic interactions served as the key factor in enhancing low-temperature catalytic oxidation activity.

Modification: Top/Page 7: It is noteworthy that the physically mixed Pt_e/γ-Al₂O₃&CeO₂-300A catalysts obtained comparable activity toward C₃H₆ and CO oxidation to Pt_e/γ-Al₂O₃-300A catalyst. It implied that the proximity mattered for Pt_e/CeO₂ catalysts, and the synergistic interactions only occurred at the Pt-O-Ce interfacial sites.

Supplementary Fig. 4 Catalytic activities of C_3H_6 and CO oxidation over Pt_e -300A and different $Pt/\gamma-Al_2O_3$ catalysts. a C_3H_6 and b CO oxidation light-off curves. Reaction condition: 1000 ppm C_3H_6 , 4000 ppm CO, and 10% O_2 in N_2 balance with a WHSV of 240, 000 $mL\ g^{-1}\ h^{-1}$. $Pt/\gamma-Al_2O_3&CeO_2$ -300A represented the physically mixed $Pt/\gamma-Al_2O_3$ and CeO_2 , which was activated by H_2 at 300 °C.

The concept of temperature-induced transitions in active sites, from metallic Pt to Pt-O-Ce interfacial sites, is a central and compelling aspect of this work. To further support the generality of this dynamic model, it would be helpful to conduct additional *in situ* characterization experiments under CO oxidation conditions. These studies could confirm whether a similar mechanistic transition occurs in other oxidation reactions, thereby reinforcing the broader applicability of the proposed model and demonstrating its relevance beyond propylene oxidation.

Response: Thank you very much for your constructive suggestion. We agree that additional *in situ* characterization using CO oxidation as a probe reaction would provide definitive proof for the feasibility of the proposed dynamic mechanism while simultaneously establishing its broader applicability. Therefore, the *in situ* Raman spectra experiments were carried out over Pt_e -300A sample under the flowing reactant composed of 4000 ppm CO and 10% O_2 balanced with N_2 at a total flow rate of 50 $mL\ min^{-1}$. As shown in the figure below (Supplementary Fig. 26), the ratio of I_D/I_{F2g} maintained at approximately 17.2% above 170 °C. However, the I_D/I_{F2g} ratio remained near 17.5% below 170 °C but abruptly decreased to 11.1% at 175 °C, followed by a progressive decline to 7.3% when the reaction temperature elevated to 225 °C. This observation confirmed that oxygen vacancy concentration remained stable below 170 °C but became activated above this threshold, indicating that the CO oxidation over Pt_e -300A catalyst also underwent a dynamic mechanistic evolution.

Modification: Middle/Page 15: Moreover, an equivalent phenomenon was detected over CO oxidation in Supplementary Fig. 26, where the I_D/I_{F2g} ratio abruptly dropped above 170 °C, confirming a similar oxygen activation process stimulated and participated in CO oxidation.

Modification: Top/Page 31: For CO oxidation, the experiments were conducted under the flowing reactant composed of 4000 ppm CO and 10% O₂ balanced with N₂ at a total flowrate of 50 mL min⁻¹.

Supplementary Fig. 26 Operando Raman spectra of Pt_c-300A under CO oxidation from 50 to 225 °C.

(3)-Supporting information

Supplementary Figure 4 lacks a legend or key explaining the meaning of the symbols used in the graph. Without this information, it is difficult for readers to interpret the data accurately. Please revise the figure or its caption to clearly define all symbols, lines, and markers used, ensuring that the figure is self-explanatory and consistent with the standards expected for supporting information.

Response: Thank you very much for your comments and suggestion. We have revised the previous Supplementary Fig. 4 by adding the legend of CeO₂ and CeO₂-300A.

Modification: Supporting information/Page S6: The revised figure was presented below. As the catalytic activity comparison of Pt-XA samples has been added as Supplementary Fig. 1, the original Supplementary Fig. 4 has been renumbered as Supplementary Fig. 5.

Supplementary Fig. 5 Catalytic activities of C₃H₆ oxidation over bare CeO₂ in the presence and absence of H₂ activation at 300 °C. Reaction condition: 1000 ppm C₃H₆, and 10% O₂ in N₂ balance with a WHSV of 240, 000 mL g⁻¹ h⁻¹.

The reported particle sizes in the text appear to be slightly inconsistent with the results presented in Supplementary Figure 21, which contains the TEM images and corresponding size distribution data. To maintain accuracy and consistency, it is recommended that the authors carefully cross-check these descriptions and ensure alignment between the written discussion and the supplementary figures. Any discrepancies should be clarified or corrected to avoid potential confusion.

Response: Thank you for your thorough review and valuable suggestions. We sincerely apologize for the inconsistency in significant figures across decimal values. To prevent potential misinterpretation, we have revised Fig. 21b to ensure precise alignment of diameter measurements with corresponding data in the supporting documentation. Additionally, Supplementary Fig. 21 has been renumbered as Supplementary Fig. 24 and is presented below.

Modification: Supporting information/Page S25: The morphology of Pt species over Pt_e/γ-Al₂O₃ and Pt_e/γ-Al₂O₃-300A catalysts was also captured and demonstrated. For Pt_e/γ-Al₂O₃ sample, Pt ensembles existed with a mean diameter of approximately 0.79 nm. Meanwhile, abundant Pt single atoms, with approximately 46%, could also be detected at the surface of Pt_e/γ-Al₂O₃ catalyst. The number of Pt single atoms was reduced dramatically after H₂ activation, accompanied by a significant rise in the average diameter of Pt ensembles (0.92 nm), which might be triggered by the migration of the Pt single atoms to the neighboring ensembles.

Supplementary Fig. 24 HAADF-STEM images and size distribution of Pt clusters over $Pt_e/\gamma\text{-Al}_2\text{O}_3$ and $Pt_e/\gamma\text{-Al}_2\text{O}_3\text{-300A}$ catalysts. a, c Additional HAADF-STEM images of $Pt_e/\gamma\text{-Al}_2\text{O}_3$ and $Pt_e/\gamma\text{-Al}_2\text{O}_3\text{-300A}$ catalysts, respectively (red cycle: Pt nanocluster; green cycle: Pt single atom); b, d size distribution of Pt ensembles over $Pt_e/\gamma\text{-Al}_2\text{O}_3$ and $Pt_e/\gamma\text{-Al}_2\text{O}_3\text{-300A}$ catalysts, respectively.

Reviewer #2 (Remarks to the Author):

This manuscript presents a comprehensive study on a hydrogen-activated Pt/CeO₂ ensemble catalyst with enhanced performance for C₃H₆ oxidation. The authors demonstrate a structural transformation from single-layer Pt to multilayer Pt ensembles upon H₂ activation and provide valuable mechanistic insights. The integration of advanced in situ characterization techniques and DFT calculations significantly reinforces the conclusions. However, several points require clarification or further elaboration to strengthen the manuscript:

Both metallic Pt⁰ and interfacial Pt–O–Ce sites are identified as important for C₃H₆ oxidation. Could the authors provide insights into the optimal ratio of these two active sites for maximizing catalytic performance?

Response: Thank you for your valuable suggestions. We have conducted catalytic oxidation of propylene over Pt_e and Pt_e-XA catalysts, which were activated by H₂ at different temperatures (Supplementary Fig. 1). The series of catalysts were further characterized by XPS to understand the Pt valence ratio (Supplementary Fig. 15). As presented in Supplementary Fig. 15a, the Pt⁰ content in Pt_e-100A, Pt_e-200A, Pt_e-300A, and Pt_e-400A was quantified as 2.3%, 44.5%, 61.0%, and 75.6%, respectively. Given that Pt_e-300A demonstrated the best low-temperature catalytic activity for C₃H₆ oxidation (Supplementary Fig. 1), the optimal ratio between Pt⁰ and Pt–O–Ce (Pt⁰/Pt^{δ+}) was expected to be 1.6.

Modification: Supporting information/Page S16: Meanwhile, since Pt_e-300A demonstrated the best low-temperature catalytic activity for both C₃H₆ and CO oxidation among Pt_e-XA samples (Supplementary Fig. 1), the optimal ratio between Pt⁰ and Pt–O–Ce was expected to be 1.6.

Supplementary Fig. 15 XPS for Pt_e-XA and Pt_e catalysts. a Pt 4f, b O 1s, and c Ce 3d regions.

Additionally, is there any specific H₂-activation temperature or duration that leads to this optimal site distribution?

Response: Thank you very much for your kind suggestion. The light-off curves of C₃H₆ oxidation over supported Pt_e catalysts, which were pretreated in 10% H₂/N₂ reducing flow under different temperatures for 1 h, have been added to the revised manuscript. As shown in the figure below (Supplementary Fig. 1), Pt_e-300A exhibited the best catalytic activities, achieving the lowest T₅₀

values compared to the supported Pt_e pretreated at 100, 200, and 400 °C (denoted as Pt_e-100A, Pt_e-200A, Pt_e-400A, respectively). These experimental results affirmed that 300 °C was the optimal activation temperature for supported Pt_e catalysts, maximizing their activities for CO and C₃H₆ oxidation reactions.

Modification: Middle/Page 26: The pretreated samples were denoted as Pt_e-XA, where X represented the activation temperature. Pt_e catalysts were pretreated at 100, 200, 300, or 400 °C to identify the optimal activation condition. As evidenced in Supplementary Fig. 1, Pt_e-300A exhibited the highest catalytic activity towards C₃H₆ oxidation among the series of Pt_e-XA catalysts.

Supplementary Fig. 1 Catalytic activities of C₃H₆ oxidation over supported Pt catalysts pretreated under different temperatures. Reaction condition: 1000 ppm C₃H₆ and 10% O₂ in N₂ balance with a WHSV of 240, 000 mL g⁻¹ h⁻¹.

In Figure 4e, the DFT analysis focuses on the hydrogen-activated multilayer Pt structure. To make a convincing argument that Pt–O–Ce alone is insufficient for C₃H₆ oxidation, the authors should also include DFT calculations for the pre-activation structure, i.e., the single-layer planar Pt configuration. While Supplementary Figure 16 presents a comparison, the electronic structure of Pt–O–Ce in a multilayer ensemble could differ significantly from that in a single-layer configuration. A direct DFT comparison of the Pt structure before and after hydrogen-activation would offer a comprehensive understanding of how structural evolution influences catalytic activity.

Response: Thank you very much for your valuable comments and suggestions. DFT calculations probing methyl group dehydrogenation in C₃H₆ over pristine Pt_e catalysts with single-layer Pt ensembles have been conducted, and the results have been summarized into Supplementary Fig. 18. As shown in the figure below (Supplementary Figs. 17 and 18), Pt⁰ sites on activated Pt_e-300A exhibited a lower activation energy barrier (1.53 eV) versus single-layer Pt ensembles on pre-activated Pt_e catalysts (1.68 eV) for the dehydrogenation process. This confirmed that metallic Pt

sites formed during H₂-triggered structural evolution were key to enhancing low-temperature C₃H₆ oxidation activity.

Modification: Bottom/Page 11: As shown in Supplementary Figs. 17-18, Pt⁰ sites on the top layers obtained a much lower energy barrier (1.53 eV) than both Pt^{δ+} at the bottom sites (2.41 eV) and single-layer Pt ensembles on pre-activated Pt_e catalysts (1.68 eV) for the dehydrogenation process, which led to the much better catalytic activity of C₃H₆ oxidation on metallic Pt sites formed during the H₂-triggered structural evolution.

Supplementary Fig. 18 DFT calculations of dehydrogenation of methyl group in C₃H₆ at the as-synthesized, supported Pt catalysts. **a** DFT calculated optimized structures of the initial state (IS), transition state (TS), final state (FS) (Grey: carbon; White: hydrogen; Blue: Pt atoms; Red: oxygen; Pale yellow: cerium); **b** Energy barriers of oxygen-facilitated dehydrogenation on single-layer Pt ensemble.

In Supplementary Figure 17, the density of states (DOS) plots for Pt⁰ and Pt^{δ+} sites appear quite similar, despite the reported d-band centers at -1.98 eV and -2.58 eV, respectively. Given the relatively small difference, this discrepancy may not be substantial enough to support definitive conclusions. To enhance the analysis, the authors should present the DOS of the propylene molecule adsorbed on both Pt⁰ and Pt^{δ+} sites, with particular attention to the HOMO structure of the carbon atoms, to better illustrate the electron-donating behavior the carbon atoms on these two types of sites.

Response: Thank you very much for your constructive suggestions. We have conducted DOS calculations regarding C₃H₆ adsorption on both Pt⁰ and Pt^{δ+} sites over Pt_e-300A catalyst. As shown in the figure below (Supplementary Fig. 21), a notable orbital hybridization between C and Pt occurred over the top site (Pt⁰) within the -5 to -10 eV energy range, with pronounced C-Pt hybridized peaks. The broad overlap across multiple energy levels indicated strong electron cloud interactions. Even though C-Pt hybridization also occurred at the side site (Pt^{δ+}), the hybridized peaks exhibited markedly reduced intensity and narrower energy distribution compared to the top configuration. Meanwhile, strong hybridization at the Pt⁰ site shifted the d-band center to lower energies to -3.02 eV, in comparison to -2.07 eV at the Pt^{δ+} site. These results demonstrated stronger interfacial interactions and more stable chemisorption at Pt⁰ sites, thereby facilitating subsequent C₃H₆ activation.

Modification: Middle/Page 12: Furthermore, DOS calculations for C₃H₆ adsorbed at Pt⁰ and Pt^{δ+} sites revealed distinct electronic interactions shown in Supplementary Fig. 21. At the Pt⁰ site, a

significant orbital hybridization between C and Pt occurred within the energy range from -5 to -10 eV. The broad overlap across multiple energy levels indicated strong electron cloud interactions. The C-Pt orbital hybridization was also present over the Pt^{δ+} site, yet the hybridized peaks exhibited markedly reduced intensity and narrower energy distribution. Meanwhile, strong hybridization at the Pt⁰ site shifted the d-band center to lower energies to -3.02 eV, in comparison to -2.07 eV at the Pt^{δ+} site. Collectively, these results demonstrated more stable C₃H₆ adsorption at top adsorption sites and stronger interfacial interactions, thereby facilitating subsequent C₃H₆ activation.

Supplementary Fig. 21 Density of states projected on the 5d-orbital of Pt and 2p-orbital of C for C₃H₆ adsorption at a Pt⁰ and b Pt^{δ+} sites. The indicated number represents the d-band center for 5d-orbital of Pt.

The manuscript lacks details regarding the transition state calculation in the “Methods” section.

Response: Thank you for your valuable comment. We have added the details of the transition state calculation in the “Methods” section. We also supplemented much more information for Pt/CeO₂ ensemble catalyst models in this section.

Modification: Top/Page 32: The transition state was located via constrained optimization based on a process of varying the target reaction coordinate while relaxing all other degrees of freedom. The optimized structure was subsequently validated by a vibrational frequency analysis confirming the presence of exactly only one imaginary frequency^{60,61}.

Reference

60 Liu, Z. P. & Hu, P. General rules for predicting where a catalytic reaction should occur on metal surfaces: a density functional theory study of C–H and C–O bond breaking/making on flat, stepped, and kinked metal surfaces, *J. Am. Chem. Soc.* **125**, 1958–1967, (2003).

61 Alavi, A., Hu, P., Deutsch, T., Silvestrelli, P.L. & Hutter, J. CO oxidation on Pt (111): an
a
b